ecology/biomathematics/applied mathematics

collective decision-making, foraging, optimality, social insects, dynamic environments

**Author for correspondence:**
Zachary P. Kilpatrick
e-mail: zpkilpat@colorado.edu

# Social inhibition maintains adaptivity and consensus of honeybees foraging in dynamic environments

Subekshya Bidari[1], Orit Peleg[2,3,4] and
Zachary P. Kilpatrick[1,5]

[1]Department of Applied Mathematics, [2]Department of Computer Science, and [3]BioFrontiers Institute, University of Colorado, Boulder, CO 80309, USA
[4]Santa Fe Institute, Santa Fe, NM 87501, USA
[5]Department of Physiology and Biophysics, University of Colorado School of Medicine, Aurora, CO 80045, USA

OP, 0000-0001-9481-7967; ZPK, 0000-0002-2835-9416

To effectively forage in natural environments, organisms must adapt to changes in the quality and yield of food sources across multiple timescales. Individuals foraging in groups act based on both their private observations and the opinions of their neighbours. How do these information sources interact in changing environments? We address this problem in the context of honeybee colonies whose inhibitory social interactions promote adaptivity and consensus needed for effective foraging. Individual and social interactions within a mathematical model of collective decisions shape the nutrition yield of a group foraging from feeders with temporally switching quality. Social interactions improve foraging from a single feeder if temporal switching is fast or feeder quality is low. When the colony chooses from multiple feeders, the most beneficial form of social interaction is direct switching, whereby bees flip the opinion of nest-mates foraging at lower-yielding feeders. Model linearization shows that effective social interactions increase the fraction of the colony at the correct feeder (consensus) and the rate at which bees reach that feeder (adaptivity). Our mathematical framework allows us to compare a suite of social inhibition mechanisms, suggesting experimental protocols for revealing effective colony foraging strategies in dynamic environments.

# 1. Introduction

Social insects forage in groups, scouting food sources and sharing information with their neighbours [1–3]. The emergent global perspective of animal collectives helps them adapt to dynamic and competitive environments in which food sources' quality and location can vary [4]. Importantly, decisions made by groups involve nonlinear interactions between individuals, temporally integrating information received from neighbours [5]. For example, honeybees *waggle dance*[1] to inform nest-mates of profitable nectar sources [6,7], and use *stop signalling*[2] to dissuade them from perilous food sources [8] or curb recruitment to overexploited sources [9]. While waggle dancing rouses bees from indecision, stop signalling prevents decision deadlock and builds consensus when two choices are of similar quality [10]. Thus, both positive and negative feedback interactions within the group are important for regulating collective decisions and foraging [11,12].

Honeybee colonies live in dynamic environments, in which the best adjacent nest or foraging sites can vary across time [13,14]. Bees adapt to change by abandoning less-profitable nectar sources for those with higher yields [15,16], and by modifying the number of foragers [17,18]. Prior studies focused on how waggle dance recruitment or the division of individual bee roles shape colony adaptivity [19,20]. Inhibitory social interactions, whereby bees stop each other from foraging, have been mostly overlooked as a communication mechanism for facilitating collective adaptation to change [21,22]. We propose that inhibitory social interactions are important for foraging groups to adapt to change in a fluid world.

To study how social inhibition shapes foraging yields, we focus on a task in which the nectar quality of feeders is switched periodically. Related situations probably occur in nature due to the dynamics of competitor and predator prevalence, crowding by nest-mates, and weather fluctuations [23–25]. Precisely periodic dynamics do not occur naturally but can be generated in controlled experiments [16,20]. There are important distinctions between the goals of colonies in foraging as opposed to those searching for a new home site. Once a colony establishes a permanent nest site, this is the starting and ending point for each food foraging excursion. The colony does not need to reach consensus to obtain nutrition from foraging, since food is brought to the nest regardless of how many foraging sites the group is split between [25]. By contrast, when a honeybee swarm looks for a nest, it must reach consensus for all bees and the queen to fly to the selected site. If not, their transition to a permanent nest site will be delayed, or the swarm might split. Bees use stop signals to obtain this needed consensus when house-hunting, especially when two potential sites are of similar quality [26]. Consensus is not essential when foraging for food, but, as we will show, increasing the fraction of the colony at the best foraging site increases foraging yields.

Foraging colonies appear to be able to adapt to change. In prior studies [15,16,20], colony foraging targets shifted in response to food quality switches, suggesting bee collectives can detect such changes. Uncommitted inspector bees can lead bees away from feeders whose nectar quality has dropped [20], and recruitment via waggle dancing appears to be unimportant for effective foraging in changing environments (see also [27]). Here, we also find recruitment can be detrimental, but social inhibition can rapidly pull bees from low- to high-yielding feeders. This, paired with 'abandonment' whereby bees spontaneously stop foraging, facilitates temporal discounting of prior evidence. By contrast, strong positive feedback via recruitment causes bees to congregate at feeders even after food quality has dropped, biasing a colony's behaviour based on past states of the world.

We quantify the contribution of these positive and negative feedback interactions within a mathematical model of a foraging colony. Our study focuses on four potential inhibitory social interactions—discriminate and indiscriminate stop signalling [8,26], direct switching [28,29] and self-inhibition—by which foraging bees alter the behaviour of other foraging bees. Self-inhibition has not been reported in honeybee foraging experiments, but we consider its effects as a potential social inhibitory mechanism, claiming it could be observable in behavioural assays for which it is advantageous (e.g. single switching feeder). Strategies are compared by measuring the rate of foraging yield over the timescale of feeder quality switches. When bees have a single feeder, social interactions are less important unless temporal switching is fast and food quality is low, but in the case of two feeders the performance of different forms of social interactions is clearly delineated. Direct switching,

---

[1]Worker bees perform this figure-eight dance after returning to the hive from foraging, indicating the direction and distance to water, high-quality flowers or potential nest sites [6].

[2]Bees direct a high-frequency body vibration at waggle dancers to try and make them stop when problems with nest or feeding sites are detected [8].

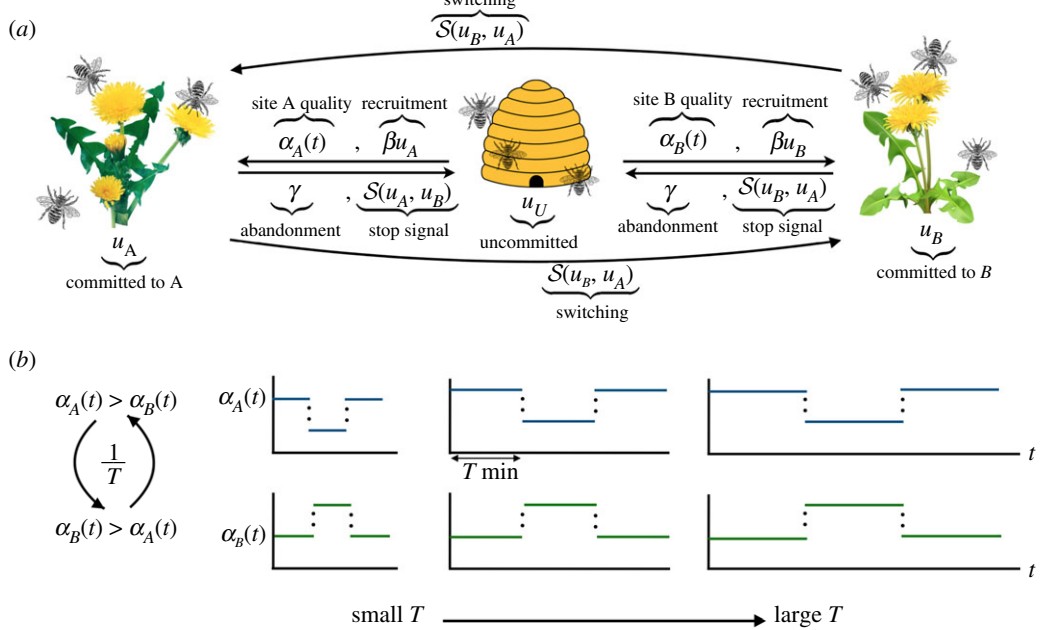

**Figure 1.** (a) Schematic of colony foraging model with two feeders (e.g. flowers or feeder boxes), equation (2.1). Bees move along arrows between different opinions (uncommitted or committed); arrow labels indicate interactions that provoke those opinion switches. (b) Example feeder quality time series $\alpha_{A,B}(t)$, which switch with period $T$ min.

by which a bee converts another forager to their own preference, is the most effective means for a colony to adapt to feeder quality changes. Also, foraging yields are most sensitive to changes in group-wide interactions in rapidly changing environments with lower food quality. Model linearizations allow us to calculate a correspondence between social interaction parameters and the *consensus* (steady-state fraction of bees at the high-yielding feeder) and *adaptivity* (the rate of switching from low- to high-yielding feeders). This provides a clear means of determining the impact of social interactions on a colony's foraging efficacy.

## 2. Results

The mathematical model of bee colony foraging decisions assumes potential foragers may be uncommitted or committed to one of the possible feeders [29]. Uncommitted bees spontaneously commit by observing a feeder or by being recruited by another currently foraging bee. Committed bees may spontaneously abandon their chosen feeder, or may be influenced to stop foraging or switch their foraging target based on inhibitory social interactions [26,29]. A population-level model emerges in the limit of large groups. Stochastic effects of the finite system do not qualitatively change our results in most cases (see appendix C(f)).

We mostly focus on two-feeder ($A$ and $B$) systems, in which the fraction of the foragers committed to either feeder is described by a pair of nonlinear differential equations in the limit of large groups (see figure 1a for a schematic)

$$\dot{u}_A = (1 - u_A - u_B)(\alpha_A(t) + \beta u_A) - \gamma u_A - \mathcal{S}(u_A, u_B) \qquad (2.1a)$$

and

$$\dot{u}_B = (1 - u_A - u_B)(\alpha_B(t) + \beta u_B) - \gamma u_B - \mathcal{S}(u_B, u_A), \qquad (2.1b)$$

where $\alpha_{A,B}(t)$ are time-dependent food qualities at feeders $A$, $B$ (see figure 1b for examples); $\beta$ min$^{-1}$ is the rate bees recruit nest-mates to their feeder via waggle dancing; $\gamma$ min$^{-1}$ is the rate bees spontaneously abandon a feeder[3]; and $\mathcal{S}(x, y)$ is a nonlinear function describing inhibitory social interactions (e.g. stop signalling or direct switching as described in appendix B(a)). Since commitment fractions are

---

[3]We have associated units of min$^{-1}$ with interaction rates. Though $\alpha_{A,B}(t)$ are in fact food qualities (see table 1 in appendix), we assume the commitment term also carries units of min$^{-1}$ via a unit rescaling, which we do not include in equation (2.1) to keep it from becoming too cumbersome. We make a similar assumption for the single-feeder model.

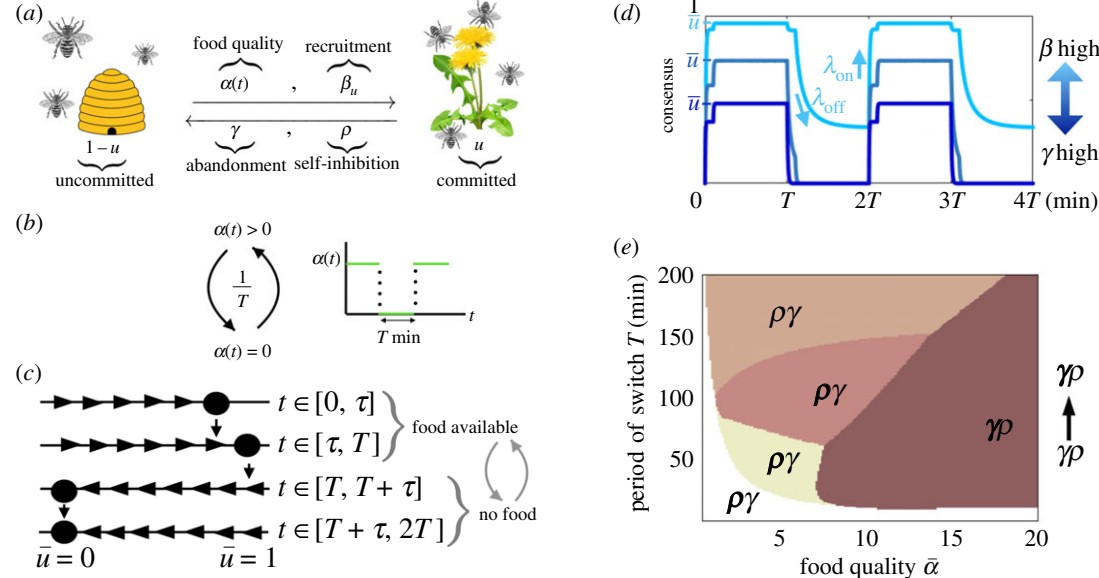

**Figure 2.** Colony dynamics in the single-feeder model. (a) Schematic of group foraging single feeder, equation (2.3). (b) Food availability $\alpha(t)$ switches on $\bar{\alpha}$ and off 0 at time intervals $T$ (min). (c) Phase line plots: equilibria of equation (2.3) within each food quality epoch are marked as dots. Dynamic increases/decreases of the foraging fraction are indicated by right/left arrows. Bees forage when food becomes available ($\alpha \to \bar{\alpha} > 0$ for $t \in [0, T)$) and $\bar{u} > 0$ is stable and abandon the feeder once food is removed ($\alpha \to 0$ for $t \in [T, 2T)$) if recruitment is weaker than abandonment ($\beta < \gamma$). (d) The fraction of bees foraging $u(t)$ tracks environmental changes. Higher/lower consensus $\bar{u}$ is obtained by changing the balance of recruitment $\beta$ and abandonment $\gamma$. (e) Reward rate maximizing strategies vary with feeder quality ($\bar{\alpha}$) and switching interval ($T$). Each coloured region denotes a different optimal strategy given the environment ($\bar{\alpha}$, $T$). The best strategies exclude recruitment ($\beta = 0$). Boldness of letters $\gamma$ and $\rho$ denote the strength of colony behaviours that best adapt to the given environment. In rapid (short $T$) or low-quality (low $\bar{\alpha}$) environments (white region), strong inhibition $\rho$ and weak abandonment $\gamma$ is best, whereas in slow or high-quality environments, inhibition $\rho$ can be weak. We take $\tau = T/10$ min throughout. See appendix A(c) for optimization methods.

bounded within the simplex $0 \le u_{A,B} \le 1$ and $0 \le u_A + u_B \le 1$, the commitment ($\alpha_{A,B}$) and recruitment ($\beta$) provide positive feedback and the abandonment ($\gamma$) and inhibition ($S$) provide negative feedback.

We assume feeders are large enough to accommodate all the bees in the colony and hence we neglect the effects of crowding. Foraging efficacy is thus quantified by the group reward rate (RR), assuming net nutrition is proportional to the fraction of the colony at a feeder $u_X$ times the current quality of that feeder minus the foraging cost $c$ (e.g. energy required to forage), $\alpha_X(t) - c$. Integrating this product and scaling by time yields the effective RR.

$$J(\alpha_{A,B}(t), \beta, \gamma, S) = \frac{1}{T_f} \int_0^{T_f} [u_A(t) \cdot (\alpha_A(t) - c) + u_B(t) \cdot (\alpha_B(t) - c)] \mathrm{d}t. \qquad (2.2)$$

Given a food quality switching schedule $\alpha_{A,B}(t)$ and total foraging time $T_f$, colonies with more efficient foraging strategies ($\beta, \gamma, S$) have higher RRs $J$.

Before studying how social inhibition shapes a bee colony foraging in two-feeder environments, we analyse the single-feeder model, finding that commitment and negative feedback from either abandonment or inhibition are usually sufficient for the group to rapidly adapt to feeder quality switches.

## 2.1. Shaping colony adaptivity and consensus for single feeders

Inhibitory social interactions in a single-feeder model can only take the form of *self-inhibition*, by which a foraging bee stops another based on a detected change in food quality (figure 2a). Since transit from the hive to the feeder takes time, we incorporate a delay of $\tau$ min, so the fraction of foraging bees $u$ evolves as

$$\dot{u} = (1 - u)(\alpha(t) + \beta u) - \gamma u - \rho(\bar{\alpha} - \alpha(t - \tau))u^2, \qquad (2.3)$$

where $\alpha(t)$ is the food quality schedule of the feeder that switches at time intervals $T$ (min) between $\alpha(t) = 0$ and $\alpha(t) = \bar{\alpha}$ [16,20] (figure 2b), $\beta$ min$^{-1}$ and $\gamma$ min$^{-1}$ are the recruitment and abandonment rates, and $\rho$ min$^{-1}$ is the rate of self-inhibition.

Colony's adaptivity and consensus is shaped by both individual behaviour changes (commitment $\alpha(t)$ and abandonment $\gamma$) and interactions (recruitment $\beta$ and inhibition $\rho$) [26]. Periodic solutions to equation (2.3) can be found explicitly, allowing us to compute a colony's RR (see appendix A(b)). Adaptive colonies rapidly return to the hive when no food is available and quickly populate the feeder when there is food (figure 2c,d). Equation (2.3) admits one stable equilibrium in each time interval: when no food is available ($\alpha(t) = 0$) the non-foraging ($\bar{u} = 0$) equilibrium is stable as long as recruitment is not stronger than abandonment ($\beta < \gamma$). When food becomes available ($\alpha(t) = \bar{\alpha} > 0$) the stable fraction of foragers $\bar{u}$ increases with food quality (see figure 2c and appendix A(a)). This fraction $\bar{u}$ corresponds to the *consensus* of the group [30], and the rate the group responds to change we deem its *adaptivity*.

### 2.1.1. Robust foraging should adapt to the environmental conditions

The performance of a colony's interaction strategies strongly depends on the feeder quality $\bar{\alpha}$ and switching time $T$. Colonies with stronger rates of abandonment $\gamma$ and self-inhibition $\rho$ more quickly leave the feeder once there is no food ($\alpha(t): \bar{\alpha} \mapsto 0$), but have limited consensus $\bar{u}$ when food becomes available ($\alpha(t): 0 \mapsto \bar{\alpha}$). Increasing the recruitment rate $\beta$, on the other hand, boosts consensus but can slow the rate at which the group abandons an empty feeder (figure 2d).

To quantify the effect of abandonment $\gamma$, recruitment $\beta$ and self-inhibition $\rho$, we compute the long-term RR of the colony, measuring the foraging yield over a single period ($2T$ min) once the group equilibrates to its periodic switching behaviour (see appendix A(b))

$$J(\gamma, \beta, \rho) = \frac{1}{2T} \int_0^{2T} (\alpha(t) - c)u(t)\mathrm{d}t, \tag{2.4}$$

where $0 < c < \bar{\alpha}$ is the cost of foraging and $\alpha(t) \in \{0, \bar{\alpha}\}$ is the quality of the feeder.

For each feeder quality level, $\bar{\alpha}$, there is an optimal foraging strategy (abandonment $\gamma$, recruitment $\beta$ and stop signalling $\rho$) within our set of possible strategies (see appendix A(c)) that maximizes the RR $J(\gamma, \beta, \rho)$ (figure 2e). Here, private information is sufficient for individual bees to commit to foraging (quality sensing $\alpha(t)$), and recruitment does not benefit the colony ($\beta = 0$). Reinforcing the majority opinion via recruitment is detrimental once the environment changes, as opposed to static environments [26,31,32]. In rapid (small $T$) or low food quality ($\bar{\alpha}$ low) environments, stronger inhibition (large $\rho$) is needed to swap group commitment when the environment changes (white region, figure 2e). This nonlinear mechanism increases the adaptivity of the group, but tempers the initial stage of consensus after the feeder is switched on (see appendix A(a) for details). On the other hand, when food is plentiful (high $\bar{\alpha}$) (brown regions, figure 2e), inhibition should be weak (small $\rho$). In intermediate environments, the best strategies interpolate these extremes.

Linearizing solutions to the model equation (2.3) provides us with a closer look at how group dynamics impact foraging yields. In sufficiently slow environments (large $T$) with small delays ($\tau \to 0$), we can linearly approximate the evolving foraging fraction (see appendix A(d))

$$u(t) \approx \begin{cases} \bar{u}(1 - e^{-\lambda_{\mathrm{on}} t}), & t \in [0, T), \\ \bar{u}e^{-\lambda_{\mathrm{off}} t}, & t \in [T, 2T), \end{cases} \tag{2.5}$$

where $\bar{u}$ is the foraging fraction (consensus) and $\lambda_{\mathrm{on/off}}$ are the rates the group arrives/departs the feeder once food is switched on/off. Plugging equation (2.5) into equation (2.4), we estimate the RR

$$J \approx \frac{\bar{u}}{2}\left[(\bar{\alpha} - c)\left(1 - \frac{1 - e^{-\lambda_{\mathrm{on}} T}}{\lambda_{\mathrm{on}} T}\right) - c\frac{1 - e^{-\lambda_{\mathrm{off}} T}}{\lambda_{\mathrm{off}} T}\right]. \tag{2.6}$$

It can be shown that $\partial_\lambda J > 0$ for $\lambda = \lambda_{\mathrm{on/off}}$, so the RR increases with the rates at which the group switches behaviours. These rates increase as abandonment $\gamma$ and social inhibition $\rho$ are strengthened (appendix A(a)). Clearly, $J$ increases with $\bar{u}$ since more bees forage when food is available. Increasing abandonment $\gamma$ tends to decrease consensus, so the most robust foraging strategies cannot use abandonment that is too rapid (appendix A(a)).

We conclude that the volatility ($1/T$) and profitability ($\bar{\alpha}$) of the environment dictate the colony interactions that yield efficient foraging strategies. One important caveat is that we bounded the interaction parameters, so group communication cannot be arbitrarily fast. This biological bound may be

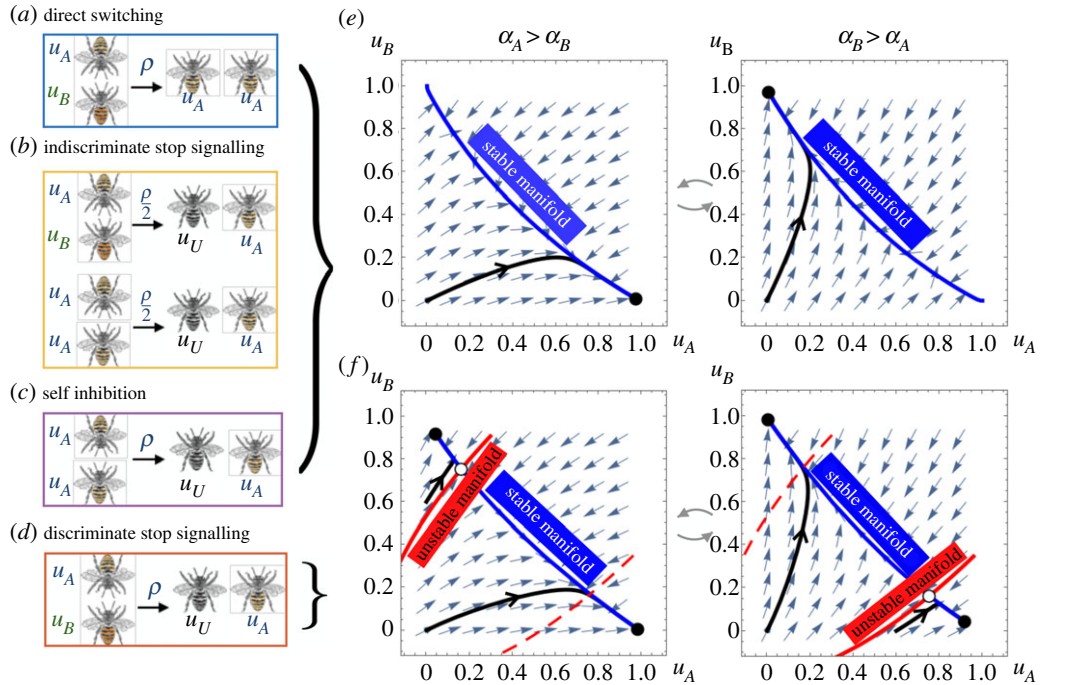

**Figure 3.** Social inhibition of colony foraging at two feeders (figure 1) and their resultant dynamical behaviours: (*a*) direct switching: inhibiting bee switches opposing nest-mate to their preference; (*b*) indiscriminate stop signalling: inhibiting bee may either cause opposing or agreeing nest-mate to become uncommitted; (*c*) self-inhibition: inhibiting bee causes agreeing nest-mate to become uncommitted; (*d*) discriminate stop signalling: inhibiting bee causes opposing nest-mate to become uncommitted. Phase portraits: (*e*) Monostable behaviour arises from direct switching, discriminate stop signalling, and self-inhibition so the group always tends to a single equilibrium foraging fraction given fixed feeder qualities. (*f*) Bistable behaviour that can emerge for strong discriminate stop signalling.

lower in practice, explaining slow adaptation of colonies to feeder changes in experiments [15,16,20]. Our qualitative finding, that social inhibition is more effective in slow and high-quality environments, should be robust to even tighter bounds. We have also shown that when social inhibition is not present, abandonment must be increased as the speed and quality of the environment is increased (appendix C(a) and figure 7). In the next section, we extend these principles to two-feeder environments, particularly showing how specific forms of social inhibition shape foraging yields.

## 2.2. Foraging decisions between two dynamic feeders

For a bee colony to effectively decide between two feeders, it must collectively inhibit foraging at the lower quality feeder. Our mean-field model, equation (2.1), generalizes house-hunting swarm models with stop signalling [10,26,32] to a foraging colony in a dynamic environment with different forms of social inhibition (figure 1). How do these inhibitory interactions contribute to foraging efficacy? Honeybees can inhibit nest-mates foraging at potentially perilous or crowded feeders [8,24,25,33], but group-level effects of these mechanisms are not well studied in dynamic environments [34]. As we will show, the specific form of social inhibition can strongly determine how a colony adapts to change.

### 2.2.1. Forms of social inhibition

Generalizing previous models [26,28], we consider four forms of social inhibition (all parametrized by $\rho$ as before): (a) direct switching: bees foraging at the superior feeder directly switch the preference of opposing foragers to the better feeder; (b) indiscriminate stop signalling: when two foraging bees meet, one will stop foraging; (c) self-inhibition: when two bees foraging at the same feeder meet, one will stop foraging; and (d) discriminate stop signalling: when bees foraging at different feeders meet, one stops foraging. These interactions are visualized in figure 3a–d and their evolution equations are given in appendix B(a) (see also electronic supplementary material of [26]).

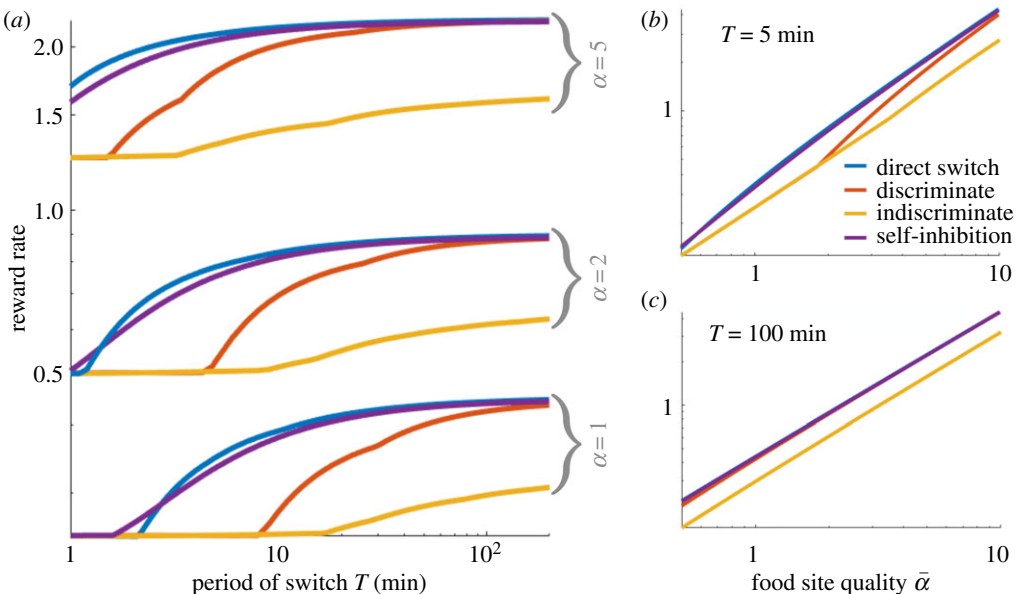

**Figure 4.** Optimal reward rates across forms of social inhibition. (*a*) Reward rate (RR) increases as the interval between feeder quality switches *T* increases for all social inhibition strategies. Across most parameter sets, direct switching is the most robust strategy, yielding the highest RRs. In rapid environments, self-inhibition can be slightly better. (*b*) For *T* = 5 min fixed and maximal food quality $\bar{\alpha}$ varied, RRs for direct switching and self-inhibition are separated from discriminate and indiscriminate stop signalling at lower food quality levels $\bar{\alpha}$. (*c*) For *T* = 100 min, direct switching, discriminate stop signalling and self-inhibition yield similar RRs, whereas indiscriminate stop signalling is notably worse. These curves are fit nearly perfectly by a linear function ($R^2 = 1.0000$).

We can divide these forms of social inhibition into two classes, based on the group foraging dynamics they produce: monostable or bistable consensus behaviours. The first three forms of social inhibition yield groups with monostable consensus behaviours (see appendix B(b) and electronic supplementary material of [26]), tending to a single stable foraging fraction when the feeder qualities are fixed (figure 3*e*). The colony will thus mostly forage at the higher-yielding feeder. On the other hand, strong discriminant stop signalling can produce colonies with bistable consensus behaviours (figure 3*f*). Such hysteresis in stop-signalling populations was also identified in [10]. As a result, the group can remain stuck at an unfavourable feeder, after the feeder qualities are switched. This is similar to 'winner-take-all' regimes in mutually inhibitory neural networks [29,35]. Inhibition from bees holding the colony's dominant preference is too strong for bees with the opposing preference to overcome, even with new evidence from the changed environment.

### 2.2.2. Direct switching leads to most robust foraging

To determine the most robust forms of social inhibition for foraging in dynamic environments, we studied how the rate of reward, equation (2.2), depended on the foraging strategy used. Environments are parametrized by the time between switches *T* (min) and the better feeder quality $\bar{\alpha}$ and the lower feeder quality $\bar{\alpha}/2$, which periodically switch between feeders *A* and *B*. As in the single-feeder case, we tune interactions of each strategy (figure 3*a–d*) to maximize RR over a discrete set of strategies (see appendix B(c) for details). Comparing each social inhibition strategy type's RR in different environments (figure 4*a*), we find direct switching generally yields higher RRs than other strategies. Differences in the effectiveness of distinct strategies are most pronounced at intermediate environmental timescales *T*. As expected, RRs increase with the maximal feeder quality $\bar{\alpha}$ (figure 4*b,c*).

Direct switching is probably a superior strategy because it allows for continual foraging (figure 3*a*), as opposed to other strategies which interrupt foraging with an uncommitted stage (figure 3*b–d*) and rely on recruitment $\beta$ to restart foraging. To study how interactions should be balanced to yield effective foraging, we examined how to optimally tune ($\beta$, $\gamma$, $\rho$) across environments in the direct switching model (figure 5). Analyses of other models are shown in figures 8 and 9 of appendix C(b).

As in the single-feeder environments, we see a delineation between strategies optimized to slow/high-quality environments as opposed to rapid/low-quality environments. Weak recruitment $\beta$ (figure 5*a*) and abandonment $\gamma$ (figure 5*b*), and strong direct switching (figure 5*c*) yield the highest

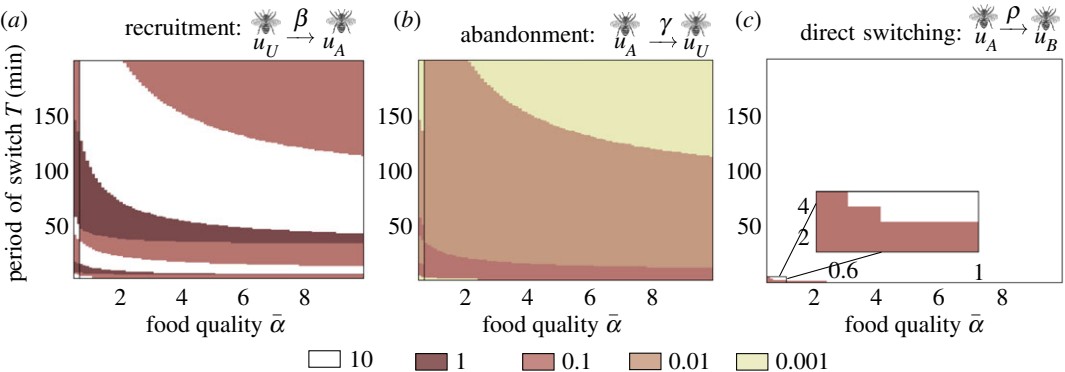

**Figure 5.** Tuning (*a*) recruitment $\beta$; (*b*) abandonment $\gamma$; and (*c*) social inhibition $\rho$ to maximize RR in the direct switching model (figure 3*a*). See appendix B(c) for methods. (*a*) The best tunings of recruitment vary considerably for rapid (low *T*) and low-quality $\bar{\alpha}$ environments, but recruitment appears to be less essential for slow (high *T*) and high-quality $\bar{\alpha}$ environments. (*b*) The rate of abandonment that best suits the environment decreases as the environment becomes slower and higher quality. (*c*) Generally, the direct switching rate should be made as strong as possible, except for very fast, low-quality environments.

RRs in slow (large *T*) and high-quality (large $\bar{\alpha}$) environments. Recruitment $\beta$ may be inessential since the food quality signals $\bar{\alpha}$ and $\bar{\alpha}/2$ are significantly different. Also, direct switching $\rho$ provides strong adaptation to change. In fact, for virtually all environments, we found it was best for $\rho$ to be as strong as possible. The strategy changes significantly when the environment is fast (small *T*) and low quality (small $\bar{\alpha}$), in which case abandonment $\gamma$ should be strong, and in extreme cases direct switching $\rho$ should be made weak (figure 5*b*,*c*). Changes in the optimal recruitment strength are less systematic, and there are stratified regions in which the best $\beta$ can change significantly for small shifts in environmental parameters. Overall, a mixture of abandonment and direct switching is more effective in more difficult environments (lower *T* and $\bar{\alpha}$).

Direct switching does underperform self-inhibition in rapid environments (figure 4*a*), since the colony can forage more efficiently by keeping some bees uncommitted, and not risking the cost of foraging at the lower-yielding feeder. Strong self-inhibition $\rho$ keeps more bees from foraging. Overall, both direct switching and self-inhibition can perform similarly, as recruitment interactions can be strengthened in self-inhibiting colonies, so more bees return to foraging after such inhibitory encounters (figure 4). This balances adaptivity, so the colony's preferences change with the environment, and consensus, so the colony mostly builds up to forage at the better feeder given sufficient time. We now study this balance in each model using linearization techniques. Overall, these measures can account for discrepancies between the RR yields of colonies using different social inhibition strategies.

### 2.2.3. Linearization reveals strategy adaptivity and consensus

Each interaction mechanism differentially shapes both the fraction of bees that forage at the better feeder in the long-time limit (consensus $\bar{u}$) and the rate at which this bound is approached (adaptivity $\lambda$). Focusing specifically on these measures, we demonstrate both how they shape foraging efficiency and how they distinguish each social inhibition strategy.

We leverage our approach developed for the single-feeder model, and consider linear approximations of equation (2.1) in the limit of long switching times *T* (see appendix B(d) and figure 10 in appendix C(c)). In the specific case $c := \bar{\alpha}/2$, we can approximate the RR solely in terms of the consensus $\bar{u}$ (long-term fraction of bees at the better feeder) and adaptivity $\lambda$ (rate this fraction is approached):

$$J \approx \frac{\bar{\alpha}}{2}\left(\bar{u} + (1 - 2\bar{u})\frac{1 - e^{-\lambda T}}{\lambda T}\right). \tag{2.7}$$

The RR *J* increases with consensus $\bar{u}$ and adaptivity $\lambda$ (figure 6*a*). Efficient colonies rapidly recruit a high fraction of the colony to the better feeder. Consensus and adaptivity are approximated in each model using linear stability (appendix B(b)). The impact of varying abandonment $\gamma$ and social inhibition $\rho$ on $\bar{u}$ and $\lambda$ is consistent with our optimality analysis of the full nonlinear model (figure 6*b*,*c*): social inhibition generates more robust switching between feeders than abandonment. While strengthening abandonment adaptivity $\gamma$ can increase $\lambda$, it decreases consensus $\bar{u}$ since it causes bees to become uncommitted (figure 6*b*). Such consensus–adaptivity trade-offs do not occur in most

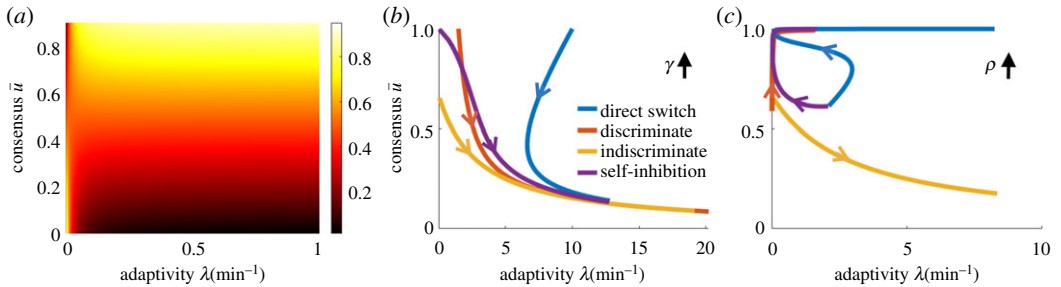

**Figure 6.** (*a*) Foraging yield varies with consensus ($\bar{u}$) and adaptivity ($\lambda$) when $T = 100$ min. Trade-off between consensus and adaptivity in the linearized model for (*b*) abandonment, $\gamma$ between [0, 20] min$^{-1}$; (*c*) social inhibition, $\rho$ between [0, 20] min$^{-1}$. Switching time $T = 100$ min, food quality $\bar{\alpha} = 2$, and other parameters are fixed at their optimal level.

models, as social inhibition $\rho$ is strengthened (see also figures 11 and 12 in appendix C(d)). Only indiscriminate stop signalling exhibits this behaviour (figure 6*c*), but the other three models (direct switching, discriminate stop signalling and self-inhibition) do not. Rather, consensus $\bar{u}$ increases with social inhibition, while adaptivity can vary nonmonotonically (direct switching) or even decrease (self-inhibition). Overall, direct switching colonies attain the highest levels of consensus and adaptivity, consistent with our finding that it is the most robust model (figure 4).

Direct switching sustains both high levels of consensus $\bar{u}$ and adaptivity $\lambda$ (figure 6*c*; see also figures 11 and 12 in appendix C(d)). The resulting colonies quickly discard their prior beliefs about the highest-yielding feeder, exhibiting leaky evidence accumulation [36]. On the other hand, strong abandonment $\gamma$ (figure 6*b*) or indiscriminate stop signalling (figure 6*c*) increase adaptivity but limit consensus $\bar{u}$ at the better feeder. Strengthening recruitment $\beta$ leads to stronger consensus $\bar{u}$ at the expense of adaptivity $\lambda$ to environmental changes. Bee colonies probably use a combination of social inhibition and abandonment mechanisms [37], which would allow flexibility in managing consensus–adaptivity trade-offs in dynamic environments.

# 3. Discussion

Foraging animals constantly encounter temporal and spatial changes in their food supply [38]. The success of foraging animal groups thus depends on how efficiently they communicate and act upon environmental changes [39]. Our bee colony model analysis pinpoints specific social inhibition mechanisms that facilitate adaptation to changes in food availability and consolidate consensus at better feeders. If bees interact by direct switching, they can immediately update their foraging preference without requiring recruitment, keeping foragers active following environmental changes. Recruitment is less important to the foraging success of a colony in dynamic conditions; bees can initiate commitment via their own scouting behaviour. Individuals should balance their social and private information in an environment-dependent way to decide and forage most efficiently [40,41].

Efficient group decision-making combines individual private evidence accumulation and information sharing across the cohort [42]. However, in groups where social influence is strong, opinions generated from weak and potentially misleading private evidence can cascade through the collective, resulting in rapid but lower value decisions [10,43,44]. Our analysis makes these notions quantitatively concrete by associating the accuracy of the colony's foraging decisions with the consensus fraction at the better feeder and the speed of decisions with the adaptivity or rate the colony approaches steady-state consensus (figure 6). The best foraging strategies balance these colony-level measures of decision efficiency. Social insects do appear to balance the speed and accuracy of decision to increase their rate of food intake [45,46], and collective tuning is probably influenced by individuals varying their response to social information.

We find that social recruitment can speed along initial foraging decisions, but it can limit adaptation to change. This is consistent with experimental studies that show a reduction in positive feedback can help collectives steer away from lower value decisions. For example, challenging environmental conditions (e.g. volatile and low food quality) are best managed by honeybee colonies whose individuals do not wait for recruitment but rely on their own individual scouting [27]. Ants encountering crowded environments tend to deposit less pheromone to keep their nest-mates from less-efficient foraging paths [47]. These experimental findings suggest social insects adapt to changing environmental conditions by limiting communication that promotes positive feedback [48]. Foragers must then be proactive in

dynamic environments, since they cannot afford to wait for new social information [49]. Thus, the advantages of social learning depend strongly on environmental conditions [50].

In concert with a reduction in recruitment, we predict that honeybee colonies foraging in volatile environments will benefit from strengthening inhibitory mechanisms at the individual and group level. Bees enacting social inhibition dissuade their nest-mates from foraging at opposing feeders. We found the most efficient form of social inhibition is direct switching whereby bees flip the opinion of committed bees to their own opinion. So do honeybees use this mechanism in dynamic environments? Observations of bee colonies making nest site decisions show scouts directly switching their dance allegiance [28,51], but these events seem to be relatively rare in the static environments of typical nest site selection experiments [31]. Other forms of social inhibitory signals, especially stop signalling, appear to be used to promote consensus in nest decisions [10,26] and escaping predators while foraging [8,24]. Thus, the role and prevalence of social inhibition as a means for foraging adaptively in dynamic environments warrants further investigation.

Most work studying the effects of social inhibition on honeybee colony decisions has focused on swarms choosing a place to build a nest from sites whose qualities are fixed in time [26,28]. Social inhibition is needed in this context to promote consensus, generating a consistent opinion across the swarm and preventing the deadlock and group splitting. On the other hand, it is not immediately obvious that social inhibition would improve foraging if it primarily increases consensus, since colonies can obtain and store food even when foragers are split between multiple feeders, though stop signals can reduce crowding [25]. Nonetheless, we found that when the colony can rapidly switch opinion so nearly all bees agree to forage from the most profitable feeder, this does increase the nutrition yield of the colony overall. However, consensus is only advantageous in dynamic environments if it does not come at a cost to adaptivity: the opinion around which consensus is built should change with the environment.

Our simple parametrized model, developed from previously validated house-hunting models [26,28,29,32], is amenable to analysis and could be validated with time-series measurements from dynamic foraging experiments. Past experimental work focused on shorter time windows in which only a few switches in feeder quality occurred [16,20], which may account for the relatively slow adaptation of the colonies to environmental changes. We predict bees will slowly tune their social learning strategies to suit the volatility of the environment, but this could require several switch observations. Foraging tasks conducted within a laboratory could be controlled to track bee interactions over long time periods using newly developed automated monitoring techniques [52]. Our study also identifies key regions in parameter space in which different foraging strategies diverge in their performance, suggesting that placing colonies in rapid environments with relatively low food supplies will help distinguish which social communication mechanisms are being used.

Previous computational modelling studies of honeybee collective decisions primarily focused on groups solving house-hunting problems in static environments [26,29], emphasizing how social interactions shape the speed at which consensus is obtained within a collective. However, less attention has been paid to how such collectives must adapt to change, and how social communication determines group adaptivity. Some previous work has discussed the importance of uncommitted inspector bees in affording group adaptivity [20], but our work is the first to systematically compare how different forms of social communication [8,26,28,29] shape group adaptivity. Social communication by which one bee can switch the foraging preference of another appear to be most effective in providing groups with the ability to both build consensus and adaptive to change. Our findings are fairly robust to considerations of interaction heterogeneity within the colony (see appendix C(e) and figure 13). A colony whose bees have individualized rates of recruitment and abandonment exhibits slight decreases in consensus and adaptivity, but qualitatively the group still remained responsive to change.

There are a number of possible extensions of our work here. For instance, one could consider separate populations of scouts and foraging recruits as in some previous modelling studies [1,53]. Our analysis assumes bees can fluidly transition between scouting and foraging behaviour, as documented in several previous studies [54,55]. Overall, a strict and unchanging division of labour within the hive provides an incomplete description of colony organization. For instance, bees may switch to foraging when the environment demands it [56] or when socially signalled to do so [57], and thus a strict caste divide between scouts and recruits may be unrealistic [55]. Honeybees' roles appear to be strongly determined by the changing requirements of the colony, such as the influx or availability of nectar, rather than strictly due to some genetic predisposition [54,58]. Bees that scout and forage tend to be in the same life cycle phase, and as such are more amenable to temporal caste switching [59]. Such flexibility may even be a rule rather than exception to colony labour organization [60]. We could also

have considered the effects of crowding at feeders [25], so nutrition yields would scale sublinearly with the fraction of bees at the feeder, possibly reordering the efficacy of social inhibition strategies.

Collective decision strategies and outcomes can depend on group size [61,62], though decision accuracy does not necessarily increase with group size [63]. We approximated bee colony dynamics using a population level model, which is the deterministic mean-field limit of a stochastic agent-based model [26]. Finite group size considerations would result in stochastic models, in which the same conditions can generate different colony dynamics [10]. The qualitative predictions of our mean-field model did not change dramatically when considering stochastic finite-size effects (see appendix C(f) and figure 14). However discriminate stop-signalling colonies exhibit bistable decision dynamics (figure 3d,f), so the stochasticity in the finite-sized model could allow colonies to break free from less-profitable feeders, similar to noise-driven escapes of particles in double potential well models [64]. Fluctuation-induced switching may thus provide an additional mechanism for flexible foraging [65,66], and would be an interesting extension of our present modelling work. Moreover, besides their importance to understanding decisions of biological collectives, our mathematical modelling results could inform efficient strategies for organizing distributed decision-making in inanimate groups, like swarm robotics and artificial communication networks [67,68].

Data accessibility. Code for producing figures is available at https://github.com/sbidari/dynamicbees
Authors' contributions. S.B., O.P. and Z.P.K. formulated scientific questions, conceived of modelling approaches, and developed models; S.B. implemented mathematical analysis and computer simulations; S.B., O.P. and Z.P.K. wrote the article. All authors gave final approval for publication.
Competing interests. We declare we have no competing interests.
Funding. S.B. and Z.P.K. were supported by NSF (grant nos. DMS-1615737 and DMS-1853630). S.B. was also supported by a Dissertation Fellowship from the American Association of University Women. Z.P.K. was also supported by NSF/NIH CRCNS (grant no. R01MH115557).
Acknowledgements. We thank Tahra Eissa for feedback on a draft of this manuscript.

# Appendix A. Colony foraging dynamics for a single switching feeder

Consider model equation (2.3) for which the food quality $\alpha(t)$ switches between two values $\alpha(t) = \bar{\alpha}$ and $0$ at length $T$ min, similar to previous experiments [16,20]. Before analysing the temporal dynamics $u(t)$ of the colony in response to food quality switches, we study equilibria and their stability to determine how different interactions within the colony impact foraging consensus and the rate at which it is approached.

## (a) Equilibrium and linear stability analysis

At any given time $t$, the dynamics of equation (2.3) are determined by the food quality function $\alpha(t)$ values at $t$ and $t - \tau$. In the time interval, $t \in [0, \tau]$, $\alpha(t) = \bar{\alpha}$ and $\alpha(t - \tau) = 0$ equilibria of equation (2.3) are solutions to

$$0 = (1 - u)(\bar{\alpha} + \beta u) - \gamma u - \rho \bar{\alpha} u^2,$$

which can be solved using the quadratic formula

$$\bar{u}_\pm^1 := \frac{1}{2}\left[\mathcal{B} \pm \sqrt{\mathcal{D}}\right], \quad \mathcal{B} = \frac{\beta - \gamma - \bar{\alpha}}{\beta + \rho\bar{\alpha}} \quad \text{and} \quad \mathcal{D} = \mathcal{B}^2 + \frac{4\bar{\alpha}}{\beta + \rho\bar{\alpha}}, \tag{A 1}$$

with linear stability given by the eigenvalues

$$\lambda_\pm^1 = \mp\sqrt{(\beta - \bar{\alpha} - \gamma)^2 + 4(\beta + \rho\bar{\alpha})\bar{\alpha}},$$

so the positive equilibrium $\bar{u}_+^1$ is stable and the negative (extraneous) equilibrium $\bar{u}_-^1$ is unstable. On $t \in [\tau, T]$, $\alpha(t) = \alpha(t - \tau) = \bar{\alpha}$ the equilibrium equation

$$0 = (1 - u)(\bar{\alpha} + \beta u) - \gamma u.$$

has solutions and eigenvalues

$$\bar{u}_\pm^2 = \frac{\beta - \bar{\alpha} - \gamma \pm \sqrt{(\beta - \bar{\alpha} - \gamma)^2 + 4\beta\bar{\alpha}}}{2\beta} \quad \text{and} \quad \lambda_\pm^2 = \mp\sqrt{(\beta - \bar{\alpha} - \gamma)^2 + 4\beta\bar{\alpha}}.$$

Again, the positive equilibrium $\bar{u}_+^2$ is stable and the negative equilibrium $\bar{u}_-^2$ is unstable. On $t \in [T, T + \tau)$, $\alpha(t) = 0$ and $\alpha(t - \tau) = \bar{\alpha}$, equilibria satisfy $0 = (1 - u)\beta u - \gamma u$, so

$$\bar{u}_0^3 = 0 \quad \text{and} \quad \bar{u}_1^3 = \frac{\beta - \gamma}{\beta},$$

and on $t \in [T + \tau, 2T)$, $\alpha(t) = \alpha(t - \tau) = 0$, so $0 = (1 - u)\beta u - \gamma u - \rho\bar{\alpha}u^2$ and

$$\bar{u}_0^4 = 0 \quad \text{and} \quad \bar{u}_1^4 = \frac{\beta - \gamma}{\beta + \rho\bar{\alpha}}.$$

Both pairs of equilibria have associated eigenvalues

$$\lambda_0 = \beta - \gamma \quad \text{and} \quad \lambda_1 = \gamma - \beta,$$

so the zero equilibria $\bar{u}_0^3 = \bar{u}_0^4 = 0$ are stable when $\gamma > \beta$ and the non-zero equilibria $\bar{u}_1^3$ and $\bar{u}_1^4$ are positive and stable when $\beta > \gamma$. Thus, to ensure no bees continue foraging when there is no food, abandonment $\gamma$ should be stronger than recruitment $\beta$.

We deem $\bar{u} := \bar{u}_+^2$ the *consensus* level, as it is the upper limit on the fraction of the bees foraging at the feeder, when it supplies food. The eigenvalues $\lambda_{\text{on}} := \lambda_+^2$ and $\lambda_{\text{off}} := \lambda_0^4$ define the *adaptivity* of the colony, or the rates of arrival to/departure from the feeder when it does/does not supply food.

## (b) Periodically forced colony foraging

Long-term periodic solutions to equation (2.3) result from switching the food quality $\alpha(t)$ between $\bar{\alpha}$ and 0 every $T$ min. These are obtained by solving equation (2.3) iteratively using separation of variables. For example, when $\alpha(t) \equiv \bar{\alpha}$ and $\alpha(t - \tau) \equiv 0$ we can separate variables and factor the resulting fraction

$$\frac{\mathrm{d}u}{u - \bar{u}_+} - \frac{\mathrm{d}u}{u - \bar{u}_-} = -(\beta + \rho\bar{\alpha})\sqrt{\mathcal{D}}\,\mathrm{d}t,$$

where $\mathcal{D}$ is defined in equation (A 1). Integrating, isolating $u$ and applying $u(0) = u_0$, we find

$$u(t) = \frac{\bar{u}_+(u_0 - \bar{u}_-) - \bar{u}_-(u_0 - \bar{u}_+)\mathrm{e}^{-(\beta+\rho\bar{\alpha})\sqrt{\mathcal{D}}t}}{u_0 - \bar{u}_- - (u_0 - \bar{u}_+)\mathrm{e}^{-(\beta+\rho\bar{\alpha})\sqrt{\mathcal{D}}t}}, \tag{A 2}$$

consistent with our equilibrium analysis showing $\lim_{t\to\infty} u(t) = \bar{u} = \bar{u}_+^2$. Now, taking $\alpha(t) \equiv \bar{\alpha}$ on $t \in [2nT, (2n + 1)T)$ for $n = 0, 1, 2, 3, \ldots$ and $\bar{\alpha} \equiv 0$ otherwise, we will have

$$\dot{u} = (1 - u)(\alpha(t) + \beta u) - \gamma u - R(t)u^2, \tag{A 3}$$

where $R(t) \equiv \rho\bar{\alpha}$ for $t \in [2nT + \tau, (2n + 1)T + \tau)$ and $R(t) \equiv 0$ otherwise. The periodic solution to equation (A 3) can be derived self-consistently by starting with an unknown initial condition $u(0) = u_0$, and then requiring $u(2T) = u_0$. Thus, within $t \in [0, \tau)$, we have the solution given by equation (A 2), and

$$u_1 := u(\tau) = \frac{\bar{u}_+(u_0 - \bar{u}_-) - \bar{u}_-(u_0 - \bar{u}_+)\mathrm{e}^{-(\beta+\rho\bar{\alpha})\sqrt{\mathcal{D}}\tau}}{u_0 - \bar{u}_- - (u_0 - \bar{u}_+)\mathrm{e}^{-(\beta+\rho\bar{\alpha})\sqrt{\mathcal{D}}\tau}}. \tag{A 4}$$

At $t = \tau$, self-inhibition vanishes and the solution is a special case of equation (A 2) for which $\rho = 0$. Thus, we can solve equation (2.3) with $u(\tau) = u_1$ as an initial condition and write for $t \in [\tau, T)$

$$u(t) = \frac{\bar{u}_+(u_1 - \bar{u}_-) - \bar{u}_-(u_1 - \bar{u}_+)\mathrm{e}^{-\beta\sqrt{\mathcal{D}}t}}{u_1 - \bar{u}_- - (u_1 - \bar{u}_+)\mathrm{e}^{-\beta\sqrt{\mathcal{D}}t}}, \tag{A 5}$$

so that at $t = T$, we have

$$u_2 := u(T) = \frac{\bar{u}_+(u_1 - \bar{u}_-) - \bar{u}_-(u_1 - \bar{u}_+)\mathrm{e}^{-\beta\sqrt{\mathcal{D}}(T-\tau)}}{u_1 - \bar{u}_- - (u_1 - \bar{u}_+)\mathrm{e}^{-\beta\sqrt{\mathcal{D}}(T-\tau)}}. \tag{A 6}$$

Beyond $t = T$, the dynamics is governed by a special case of equation (A 5) for which $(\bar{u}_+, \bar{u}_-) = (1 - (\gamma/\beta), 0)$ if $\beta > \gamma$ and $(\bar{u}_+, \bar{u}_-) = (0, 1 - (\gamma/\beta))$ if $\beta < \gamma$, so on $t \in [T, T + \tau)$:

$$u(t) = \frac{u_2(\beta - \gamma)}{\beta u_2 - (\beta u_2 + \gamma - \beta)\mathrm{e}^{(\gamma-\beta)t}},$$

for $\beta \neq \gamma$, and the limit as $\gamma \to \beta$ is $u(t) = u_2/(1 + u_2\beta t)$, which can both be evaluated at $t = T + \tau$ to yield

$$u_3 := u(T + \tau) = \begin{cases} \frac{u_2(\beta-\gamma)}{\beta u_2 - (\beta u_2 + \gamma - \beta)e^{(\gamma-\beta)\tau}} & : \beta \neq \gamma \\ \frac{u_2}{1+u_2\beta\tau} & : \beta = \gamma. \end{cases} \tag{A 7}$$

At $t = T + \tau$, self-inhibition returns since $\alpha(t - \tau) \equiv 0$, increases the negative feedback acting on foragers. The long-term steady state is determined by the balance of abandonment and recruitment: $(\bar{u}_+, \bar{u}_-) = (\beta - \gamma/\beta + \rho\bar{\alpha}, 0)$ if $\beta > \gamma$ and $(\bar{u}_+, \bar{u}_-) = (0, \beta - \gamma/\beta + \rho\bar{\alpha})$ if $\beta < \gamma$. Thus,

$$u(t) = \frac{u_3(\beta - \gamma)}{(\beta + \rho\bar{\alpha})u_3 - ((\beta + \rho\bar{\alpha})u_3 + \gamma - \beta)e^{(\gamma-\beta)t}}$$

for $\beta \neq \gamma$, and in the limit $\beta \to \gamma$, $u(t) = u_3/(1 + u_3(\beta + \rho\bar{\alpha})t)$. Both expressions can be evaluated at $t = 2T$, and self-consistency of the periodic solution requires $u_4 \equiv u_0$,

$$u_0 = u_4 := u(2T) = \begin{cases} \frac{u_3(\beta-\gamma)}{(\beta+\rho\bar{\alpha})u_3 - ((\beta+\rho\bar{\alpha})u_3+\gamma-\beta)e^{(\gamma-\beta)(T-\tau)}} & : \beta \neq \gamma \\ \frac{u_3}{1+u_3(\beta+\rho\bar{\alpha})(T-\tau)} & : \beta = \gamma \end{cases} \tag{A 8}$$

equations (A 4), (A 6), (A 7) and (A 8) can be solved explicitly for $(u_0, u_1, u_2, u_3)$, although the expressions are quite cumbersome, so we omit them here. These analytic solution techniques were used to generate the foraging fraction trajectories plotted in figure 2d and to identify model parameter that optimize the RR $J$ in different environments (plotted in figure 2e) as we now describe.

## (c) Optimizing reward rate over strategy sets

We optimized the RR of the colony foraging a single switching feeder by restricting the strategies to a discrete set of interaction parameter values. The RR in large regions of parameter space was relatively flat since it involves the sum of several exponentially small terms. To avoid spurious convergence, we focused on each parameter's relevant order of magnitude which led to the highest long-term RR. For a given environment $(\alpha, T)$, we identified the combination of interaction parameter values from the set $(\beta, \gamma, \rho) \in \{0.01, 0.1, 1, 10\}^3$ (in min$^{-1}$) yielding the highest RR $J$ computed from equation (2.4). Bounds on interaction parameters were imposed so that a colony could not completely dispense with any interaction or feedback mechanism or strengthen any to be arbitrarily rapid. This was performed over a mesh of environmental parameters $\bar{\alpha} \in [0.5, 20]$ (at $\Delta\bar{\alpha} = 0.1$ steps) and $T \in [1, 200]$ (at $\Delta T = 1$ min). We found that $\beta = 0.01$ min$^{-1}$ was optimal across all environment types, but that $\gamma$ and $\rho$ varied in strength dependent on the environmental conditions (figure 2e).

## (d) Linear approximation of the periodic solution and reward rate

The RR equation (2.4) for the single feeder can be estimated by linearly approximating the colony dynamics using results from our equilibrium analysis. Assuming the interval $T$ and between feeder quality switches ($\alpha: \bar{\alpha} \mapsto 0$; $\alpha: 0 \mapsto \bar{\alpha}$) and the delay $\tau$ are large, the colony will nearly equilibrate before each switch, suggesting the following linear approximation of the foraging fraction:

$$u(t) = \begin{cases} \bar{u}^1 + e^{-\lambda^1 t}(\bar{u}^4 - \bar{u}^1), & t \in [0, \tau] \\ \bar{u}^2 + e^{-\lambda^2(t-\tau)}(\bar{u}^1 - \bar{u}^2), & t \in [\tau, T] \\ \bar{u}^3 + e^{-\lambda^3(t-T)}(\bar{u}^2 - \bar{u}^3), & t \in [T, T+\tau] \\ \bar{u}^4 + e^{-\lambda^4(t-T-\tau)}(\bar{u}^3 - \bar{u}^4), & t \in [T+\tau, 2T], \end{cases}$$

where $\bar{u}^i$ are the stable equilibria and $\bar{u}^3 = \bar{u}^4 = 0$ when $\beta < \gamma$. Considering this case, we can compute the RR using the single-feeder version of equation (2.2) in the long-time limit

$$J = \frac{1}{2T} \int_0^{2T} u(t)(\alpha(t) - c)\,dt$$

$$= \frac{\bar{\alpha} - c}{2T} \int_0^\tau \bar{u}^1(1 - e^{-\lambda^1 t})\,dt + \frac{\bar{\alpha} - c}{2T} \int_0^{T-\tau} (\bar{u}^2 + e^{-\lambda^2 t}(\bar{u}^1 - \bar{u}^2))\,dt - \frac{c}{2T} \int_0^\tau \bar{u}^2 e^{-\lambda^3 t}\,dt$$

$$= \frac{\bar{\alpha} - c}{2T} \left( \bar{u}^1\tau - \bar{u}^1 \frac{1 - e^{-\lambda^1 \tau}}{\lambda^1} \right) + \frac{\bar{\alpha} - c}{2T} \left( \bar{u}^2(T - \tau) + \frac{\bar{u}^1 - \bar{u}^2}{\lambda^2}(1 - e^{-\lambda^2(T-\tau)}) \right)$$

$$- \frac{c}{2T} \left( \frac{\bar{u}^2}{\lambda^3}(1 - e^{-\lambda^3 \tau}) \right).$$

In the long interval $\lim_{T\to\infty}$ and short delay $\lim_{\tau\to0}$ (omitting the intermediate delay equilibria) limits, we can simplify the expression as

$$J = \frac{\bar{u}}{2}\left[(\bar{\alpha}-c)\left(1 - \frac{1-e^{-\lambda_{\mathrm{on}}T}}{\lambda_{\mathrm{on}}T}\right) - c\frac{1-e^{-\lambda_{\mathrm{off}}T}}{\lambda_{\mathrm{off}}T}\right],$$

where $\bar{u} = \bar{u}^2$, $\lambda_{\mathrm{on}} = \lambda^2 = \lambda_+^2$ and $\lambda_{\mathrm{off}} = \lambda^4 = \lambda_0$, as written in equation (2.6). For the specific case in which $\bar{\alpha} = 2$ and $c = 1$, we can write this more cleanly as

$$J(\alpha(t), \beta, \gamma, \rho) = \frac{\bar{u}}{2}\left[\left(1 - \frac{1-e^{-\lambda_{\mathrm{on}}T}}{\lambda_{\mathrm{on}}T}\right) - \frac{1-e^{-\lambda_{\mathrm{off}}T}}{\lambda_{\mathrm{off}}T}\right].$$

Clearly, increasing consensus ($\bar{u}$) and adaptivity ($\lambda_{\mathrm{on/off}}$) increases the RR.

As $\beta \to 0$, $\lambda_{\mathrm{off}} = -\gamma$, $\bar{u} = 2/[2+\gamma]$, with $\lambda_{\mathrm{on}} = -(2+\gamma)$. Increasing the rate of abandonment $\gamma$ decreases consensus $\bar{u}$ but will increase the adaptivity of the colony as both $\lambda_{\mathrm{off}} = -\gamma$ and $\lambda_{\mathrm{off}} = -(\gamma+2)$ increase in amplitude. Optimizing the RR then requires balancing these two effects. Identifying the $\gamma$ value that maximizes the RR can then be done by finding the maximum of

$$J(\alpha(t), 0, \gamma, \rho) = \frac{1}{2+\gamma}\left[\left(1 - \frac{1-e^{-(2+\gamma)T}}{(2+\gamma)T}\right) - \frac{1-e^{-\gamma T}}{\gamma T}\right],$$

given by the $\gamma$ solving $\partial_\gamma J(\alpha(t), 0, \gamma, \rho) = 0$. This analysis can be extended to consider the solutions to the full nonlinear equations, but the general trends are the same. Increasing negative feedback will tend to limit consensus while making the colony more adaptive to change.

# Appendix B. Dynamics of colonies foraging at two switching feeders

Here, we provide more details and analysis on our colony model equation (2.1) foraging between two feeders. As in the single-feeder model, we can leverage equilibria, stability and linearization to better understand the impact of model tuning on the RRs of the foraging collective.

## (a) Forms of social inhibitions

Here, we provide detailed descriptions of the dynamical models associated with each type of social inhibition used in the model of a bee colony foraging two feeders, as generalized in equation (2.1). In the main text, we simply indicate the general form of social inhibition with the function $\mathcal{S}(u_A, u_B)$, but we provide the functional form for these interactions in the descriptions below.

*Direct switching model.* A bee committed to a feeder inhibits bees with opposing opinions by causing them to switch the feeder to which they are committed

$$\dot{u}_A = (1 - u_A - u_B)(\alpha_A(t) + \beta u_A) - \gamma u_A - \rho(\alpha_B(t-\tau) - \alpha_A(t-\tau))u_A u_B$$

and

$$\dot{u}_B = (1 - u_A - u_B)(\alpha_B(t) + \beta u_B) - \gamma u_B - \rho(\alpha_A(t-\tau) - \alpha_B(t-\tau))u_A u_B,$$

where $\tau$ (in minutes) indicates the time delay required for the strength of the direct switching signal (based on detected food quality) to update following a switch in the food quality. Note that the social inhibition terms in either evolution equation ($u_A$ or $u_B$) will necessarily be of opposite sign since each is the negative of the other and $\alpha_A(t) \neq \alpha_B(t)$ for all $t > 0$.

*Indiscriminate stop-signal model.* A bee committed to a feeder indiscriminately inhibits bees committed to either feeder, affecting both bees committed to the same feeder and those committed to a different feeder, and causes them become uncommitted

$$\dot{u}_A = (1 - u_A - u_B)(\alpha_A(t) + \beta u_A) - \gamma u_A - \frac{1}{2}\rho(\alpha_A(t-\tau)u_A^2 + \alpha_B(t-\tau)u_A u_B)$$

and

$$\dot{u}_B = (1 - u_A - u_B)(\alpha_B(t) + \beta u_B) - \gamma u_B - \frac{1}{2}\rho(\alpha_B(t-\tau)u_B^2 + \alpha_A(t-\tau)u_A u_B),$$

where $\tau$ (in minutes) is the time delay required for food quality switch detection as in the direct switching model. This form of social inhibition will always lead to negative feedback to both populations as long as $\rho > 0$.

*Self-inhibition model.* A bee committed to one feeder inhibits bees committed to the same feeder, causing them to become uncommitted

$$\dot{u}_A = (1 - u_A - u_B)(\alpha_A(t) + \beta u_A) - \gamma u_A - \rho(\bar{\alpha} - \alpha_A(t - \tau))u_A^2$$

and

$$\dot{u}_B = (1 - u_A - u_B)(\alpha_B(t) + \beta u_B) - \gamma u_B - \rho(\bar{\alpha} - \alpha_B(t - \tau))u_B^2,$$

where $\tau$ is the time delay. Self-inhibition is only active in the population of foragers for which the colony detects there is less than the maximum supply of food available ($\alpha_{A,B}(t - \tau) < \bar{\alpha}$).

*Discriminate stop signal.* A bee committed to a feeder inhibits bees committed to different feeders, causing them to become uncommitted

$$\dot{u}_A = (1 - u_A - u_B)(\alpha_A(t) + \beta u_A) - \gamma u_A - \rho\alpha_B(t - \tau)u_A u_B$$

and

$$\dot{u}_B = (1 - u_A - u_B)(\alpha_B(t) + \beta u_B) - \gamma u_B - \rho\alpha_A(t - \tau)u_A u_B,$$

where $\tau$ is the time delay. The strength of the stop signal varies with the detected quality of each feeder.

## (b) Equilibria and linear stability

We determined linear approximations of periodic solutions to the full nonlinear model equation (2.1) by studying the equilibria and linear stability properties of the full system. For any $t$, the dynamics of equation (2.1) is governed by the piecewise constant values of the food quality functions denoted $\bar{\alpha}_{A,B}^t = \alpha_{A,B}(t)$ and $\bar{\alpha}_{A,B}^\tau = \alpha_{A,B}(t - \tau)$ as the actual current and delay-observed values so that

$$0 = (1 - \bar{u}_A - \bar{u}_B)(\bar{\alpha}_A^t + \beta\bar{u}_A) - \gamma\bar{u}_A - \mathcal{S}(\bar{u}_A, \bar{u}_B; \rho, \bar{\alpha}_A^\tau, \bar{\alpha}_B^\tau) \qquad \text{(C 1a)}$$

and

$$0 = (1 - \bar{u}_A - \bar{u}_B)(\bar{\alpha}_B^t + \beta\bar{u}_B) - \gamma\bar{u}_B - \mathcal{S}(\bar{u}_B, \bar{u}_A; \rho, \bar{\alpha}_B^\tau, \bar{\alpha}_A^\tau), \qquad \text{(C 1b)}$$

where $\mathcal{S}(x, y; \alpha_x, \alpha_y)$ is the nonlinear function describing inhibitory social interactions, parametrized by the strength $\rho$ and delayed quality observations $\bar{\alpha}_{A,B}^\tau$ (see §3 for exact forms). Since equation (B 1) is autonomous for piecewise time intervals, equilibria can be defined on each interval [69]. Equation (B 1) can be explicitly solved using the quartic equation for all models and time intervals, but the expressions for $\bar{u}_{A,B}$ are unwieldy, so we do not write them here.

Linear stability was classified using the eigenvalues $\lambda_\pm = 1/2[\text{Tr}(\mathcal{D}) \pm \sqrt{\text{Tr}(\mathcal{D})^2 - 4\det(\mathcal{D})}]$ of the Jacobian about fixed points $(u_A, u_B) = (\bar{u}_A, \bar{u}_B)$,

$$\mathcal{D} = \begin{bmatrix} -\bar{\alpha} + \beta(1 - \bar{u}_B - 2\bar{u}_A) - \gamma - \partial_{u_A}\mathcal{S}(\bar{u}_A, \bar{u}_B) & -\bar{\alpha} - \beta\bar{u}_A - \partial_{u_B}\mathcal{S}(\bar{u}_A, \bar{u}_B) \\ -\frac{\bar{\alpha}}{2} - \beta\bar{u}_B - \partial_{u_A}\mathcal{S}(\bar{u}_B, \bar{u}_A) & -\frac{\bar{\alpha}}{2} + \beta(1 - \bar{u}_A - 2\bar{u}_B) - \gamma - \partial_{u_B}\mathcal{S}(\bar{u}_B, \bar{u}_A) \end{bmatrix}.$$

Specific cases are stable nodes (with two negative real eigenvalues, $\lambda_\pm < 0$) and saddles (with one negative/one positive real eigenvalue, $\lambda_\pm \gtrless 0$) as illustrated figure 3. Direct switching, self-inhibition and indiscriminate stop signalling yield monostable behaviour—a phase space only containing a single stable node (figure 3e)—and the majority of bees forage at the high-yielding feeder ($\bar{u}_A > \bar{u}_B$).

Discriminate stop-signalling model can generate bistability for weak abandonment $\gamma$ and strong recruitment $\beta$ and stop signalling $\rho$. In this case, the phase space is occupied by two stable nodes separated by a saddle point (figure 3f). In this case, consensus is lower in some phases of the foraging cycle, since discriminate stop signalling prevents switching between feeders.

## (c) Optimizing reward rate over strategy sets

As in the one-feeder case, we identified the set $(\beta, \gamma, \rho) \in \{0.001, 0.1, 1, 10\}^3$ (min⁻¹) yielding the highest RR from equation (2.2) in each environment ($\bar{\alpha}, T$). For each parametrized form of social inhibition, we numerically found periodic solutions to equation (2.1) taking $\alpha_A = \bar{\alpha}$ and $\alpha_B = \bar{\alpha}/2$ initially and flipping the feeder qualities every $T$ min. This was performed over a mesh of environmental parameters

$\bar{\alpha} \in [0.5, 20]$ (at $\Delta\bar{\alpha} = 0.1$ steps) and $T \in [1, 200]$ (at $\Delta T = 1$ min). See figure 5 for direct switching and figures 8 and 9 for discriminate and indiscriminate stop-signalling model, respectively. For the self-inhibition model, the optimal strategy is low abandonment ($\gamma = 0.01$ min$^{-1}$) with high recruitment and social inhibition ($\beta = \rho = 10$ min$^{-1}$). The maximum RR is plotted for each of the four models in a given environmental condition (food quality $\bar{\alpha}$ and switching period $T$) in figure 4.

## (d) Linear approximation of the periodic solution

To compute consensus and adaptivity, we derived a linear approximation to the period solution in the two-feeder case. Feeder qualities started with $\alpha_A(t) = \bar{\alpha}$ and $\alpha_B(t) = \bar{\alpha}/2$ and switched every $T$ min. Assuming $T$ large, the colony will equilibrate between condition switching, yielding the following estimate of the $u_A(t)$ part of the periodic solution

$$u_A(t) = \begin{cases} \bar{u}^1 + e^{-\lambda^1 t}(\bar{u}^4 - \bar{u}^1), & t \in [0, \tau] \\ \bar{u}^2 + e^{-\lambda^2(t-\tau)}(\bar{u}^1 - \bar{u}^2), & t \in [\tau, T] \\ \bar{u}^3 + e^{-\lambda^3(t-T)}(\bar{u}^2 - \bar{u}^3), & t \in [T, T+\tau] \\ \bar{u}^4 + e^{-\lambda^4(t-T-\tau)}(\bar{u}^3 - \bar{u}^4), & t \in [T+\tau, 2T], \end{cases}$$

where $\bar{u}^i$ are stable equilibria in each time interval and $\lambda^j$ are the least-negative associated eigenvalues determining the decay rate to the fixed point. There is a similar expression for the opposing feeder population, $u_B(t) = u_A(t+T)$. This implies $\lambda^1 = \lambda^3$ and $\lambda^2 = \lambda^4$. In the long-time limit, the RR equation (2.2) can be computed

$$\begin{aligned} J &= \frac{1}{T_f} \int_0^{T_f} [u_A(t) \cdot (\alpha_A(t) - c) + u_B(t) \cdot (\alpha_B(t) - c)]\mathrm{d}t \\ &= \frac{\bar{\alpha} - c}{2T} \int_0^\tau (\bar{u}^1 + e^{-\lambda^1 t}(\bar{u}^4 - \bar{u}^1))\mathrm{d}t + \frac{\bar{\alpha}/2 - c}{2T} \int_0^\tau (\bar{u}^3 + e^{-\lambda^1 t}(\bar{u}^2 - \bar{u}^3))\mathrm{d}t \\ &\quad + \frac{\bar{\alpha} - c}{2T} \int_0^{T-\tau} (\bar{u}^2 + e^{-\lambda^2 t}(\bar{u}^1 - \bar{u}^2))\mathrm{d}t + \frac{\bar{\alpha}/2 - c}{2T} \int_0^{T-\tau} (\bar{u}^4 + e^{-\lambda^2 t}(\bar{u}^3 - \bar{u}^4))\mathrm{d}t \\ &\quad + \frac{\bar{\alpha}/2 - c}{2T} \int_0^\tau (\bar{u}^3 + e^{-\lambda^1 t}(\bar{u}^2 - \bar{u}^3))\mathrm{d}t + \frac{\bar{\alpha} - c}{2T} \int_0^\tau (\bar{u}^1 + e^{-\lambda^1 t}(\bar{u}^4 - \bar{u}^1))\mathrm{d}t \\ &\quad + \frac{\bar{\alpha}/2 - c}{2T} \int_0^{T-\tau} (\bar{u}^4 + e^{-\lambda^2 t}(\bar{u}^3 - \bar{u}^4))\mathrm{d}t + \frac{\bar{\alpha} - c}{2T} \int_0^{T-\tau} (\bar{u}^2 + e^{-\lambda^2 t}(\bar{u}^1 - \bar{u}^2))\mathrm{d}t \\ &= \frac{\bar{\alpha} - c}{T} \left[ \left( \bar{u}^1 \tau + \frac{\bar{u}^4 - \bar{u}^1}{\lambda^1}(1 - e^{-\lambda^1 \tau}) \right) + \left( \bar{u}^2(T - \tau) + \frac{\bar{u}^1 - \bar{u}^2}{\lambda^2}(1 - e^{-\lambda^2(T-\tau)}) \right) \right] \\ &\quad + \frac{\bar{\alpha}/2 - c}{T} \left[ \left( \bar{u}^3 \tau + \frac{\bar{u}^2 - \bar{u}^3}{\lambda^3}(1 - e^{-\lambda^3 \tau}) \right) + \left( \bar{u}^4(T - \tau) + \frac{\bar{u}^3 - \bar{u}^4}{\lambda^4}(1 - e^{-\lambda^4(T-\tau)}) \right) \right]. \end{aligned}$$

Considering the limit of long time intervals $\lim_{T\to\infty}$ and short delays $\lim_{\tau\to 0}$ and the case in which $\bar{u}^2 + \bar{u}^4 \approx 1$ (no uncommitted bees in the long-time limit) we further simplify the expression

$$J = (\bar{\alpha} - c)\left[ \bar{u}^2 + (1 - 2\bar{u}^2)\frac{1 - e^{-\lambda^2 T}}{\lambda^2 T} \right] + \left( \frac{\bar{\alpha}}{2} - c \right)\left[ 1 - \bar{u}^2 + (2\bar{u}^2 - 1)\frac{1 - e^{-\lambda^4 T}}{\lambda^4 T} \right].$$

For the specific case in which $c = \bar{\alpha}/2$, we remove the superscripts so $\bar{u} = \bar{u}^2$ and $\lambda = \lambda^2$

$$J = \frac{\bar{\alpha}}{2}\left( \bar{u} + (1 - 2\bar{u})\frac{1 - e^{-\lambda T}}{\lambda T} \right).$$

The gradient of the RR along $\bar{u}$ and $\lambda$ can then be computed as

$$\partial_{\bar{u}} J = \frac{\bar{\alpha}}{2}\left( 1 - 2\frac{1 - e^{-\lambda T}}{\lambda T} \right) \quad \text{and} \quad \partial_\lambda J = (2\bar{u} - 1)\frac{\bar{\alpha} e^{-\lambda T}}{2\lambda^2 T}(e^{\lambda T} - \lambda T - 1),$$

showing $J$ is increasing in $\bar{u}$ as long as $\lambda T > 1.594$ and increasing in $\lambda$ as long as $\bar{u} > 0.5$.

**Table 1.** Model parameters for single-feeder equation (2.3) and two-feeder equation (2.1) foraging models.

| parameter | description | numerical range | citation |
|---|---|---|---|
| $\bar{\alpha}$ | quality of food source | $[0.5, 20]$ M(mol l$^{-1}$) | [7,20] |
| $\beta$ | recruitment rate | $\mathcal{O}(10^{-1} - 10^{1})$ min$^{-1}$ | [1,26] supplement |
| $\gamma$ | abandonment rate | $\mathcal{O}(10^{-2} - 10^{1})$ min$^{-1}$ | [26] supplement |
| $\rho$ | rate of social inhibition | $\mathcal{O}(10^{-1} - 10^{1})$ min$^{-1}$ | [26] supplement |
| $T$ | period of environment switch | 1–200 min | [20] |
| $\tau$ | time delay for switch to be sensed | $0.1 \times T$ min | — |
| $c$ | cost of foraging | $\frac{\bar{\alpha}}{2}$ | — |

# Appendix C. Supplemental figures and table

## (a) Matching abandonment rate to switching rate in a single dynamic feeder

Considering the single-feeder foraging colony model without nonlinear negative feedback ($\rho = 0$ so delays $\tau$ are irrelevant), we can explicitly compute the RR $J$ as a function of other parameters. We found that the best strategies do not use recruitment ($\beta = 0$), so the abandonment rate $\gamma$ is the only parameter that needs to be tuned with the environmental switching time $T$ and food quality $\bar{\alpha}$.

Thus, equation (2.3) is linear and so the linear approximation of the periodic solution is exact, described by

$$u(t) = \begin{cases} A + (u_0 - A)e^{-(\bar{\alpha}+\gamma)t}, & t \in [0, T) \\ u_1 e^{-\gamma(t-T)}, & t \in [T, 2T), \end{cases}$$

where $u_0 = A(1 - e^{-(\bar{\alpha}+\gamma)T})/(e^{\gamma T} - e^{-(\bar{\alpha}+\gamma)T})$ and $u_1 = A(e^{\gamma T} - e^{-\bar{\alpha}T})/(e^{\gamma T} - e^{-(\bar{\alpha}+\gamma)T})$. As such, we can explicitly compute the RR equation (2.4),

$$J = \frac{\bar{\alpha} - c}{2T}\left(AT + \frac{u_0 - A}{\bar{\alpha} + \gamma}(1 - e^{-(\bar{\alpha}+\gamma)T})\right) - \frac{c}{2T}\left(\frac{A}{\gamma}(1 - e^{-\gamma T})\right),$$

determining the maximum with respect to the abandonment rate $\gamma$ by solving $\partial_\gamma J = 0$ (figure 7). The optimal abandonment rate $\gamma$ increases with the feeder quality and decreases with the switching time $T$ of the environment. Thus, the negative feedback process should adapt to the dynamics of the environment, and discounting can be more rapid when the evidence for feeder quality is stronger.

## (b) Foraging strategies with discriminate and indiscriminate stop signalling

Similar to figure 5, we optimized interactions for the discriminate stop-signalling and indiscriminate stop-signalling model to yield the highest RR equation (2.2).

In the discriminate stop-signalling model, weak recruitment $\beta$ (figure 8a), weak abandonment $\gamma$ (figure 8b), and strong stop signalling (figure 8c) yield the highest RRs for most environments ($\bar{\alpha}, T$). In slow (large $T$) and high-quality $\bar{\alpha}$ environments, abandonment $\gamma$ should be strong, and discriminate stop signalling $\rho$ can be made weak (figure 8b,c). Recruitment should be weak in most environments (figure 8a).

There is no clear preferred interaction profile for maximizing RR across environments ($\bar{\alpha}, T$) in the case of indiscriminate stop signalling (figure 9). Interestingly, the strength of indiscriminate stop-signalling parameter $\rho$ should be made low for virtually all environments (figure 9c), and thus it does not seem to improve foraging efficiency. Consensus is lower due to the non-selectivity of social inhibition to all foraging bees.

For the self-inhibition model, to maximize foraging efficiency, abandonment should be made weak ($\gamma = 0.01$ min$^{-1}$) while recruitment and social inhibition should be made strong ($\beta = \rho = 10$ min$^{-1}$).

## (c) Accuracy of linear approximations of periodic solutions

Linear approximations of the periodic solutions to the single-feeder equation (2.3) and two-feeder equation (2.1) match the evolution of the full models across a wide range of parameters and forms of

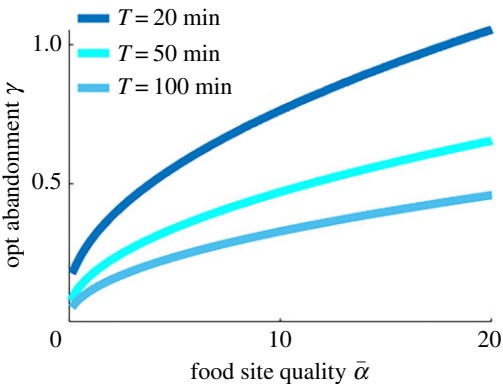

**Figure 7.** Reward rate, equation (2.4), maximizing values of abandonment ($\gamma$) parameter for a given food quality ($\alpha$) and switching period ($T$) in the single-feeder model. Colony can maximize the reward $J$ by calibrating the level of abandonment with the switching rate and feeder quality, discounting faster as the environment changes more quickly.

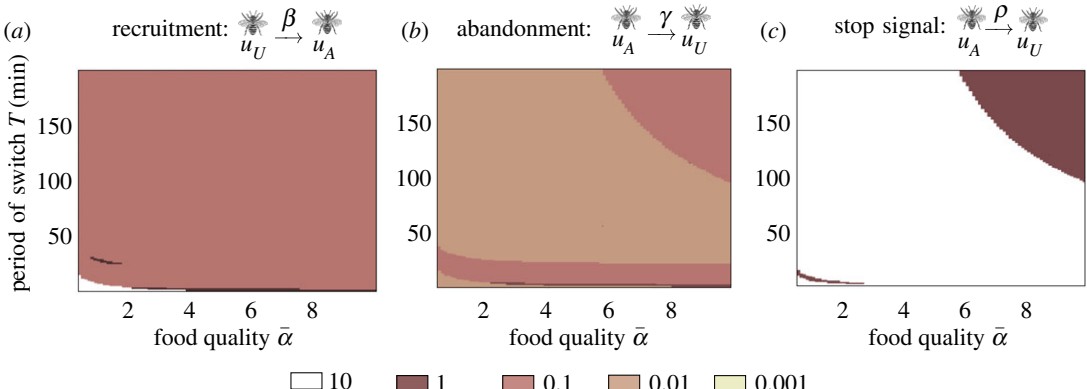

**Figure 8.** Tuning (a) recruitment $\beta$; (b) abandonment $\gamma$; and (c) social inhibition $\rho$ to maximize the reward rate (RR) (equation (2.2)), in the discriminate stop-signalling model. (a) Recruitment and (b) abandonment should be made weak whereas (c) social inhibition should be made strong except for in slow (high $T$) and high-quality $\bar{\alpha}$ environments.

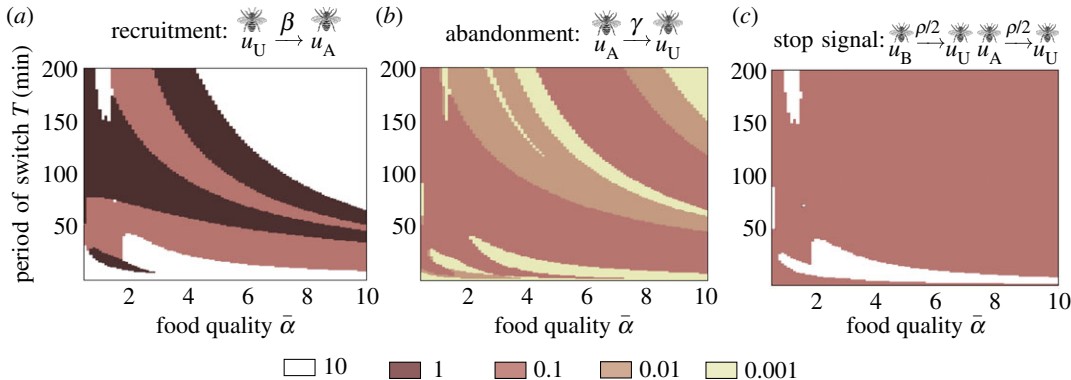

**Figure 9.** Tuning (a) recruitment $\beta$; (b) abandonment $\gamma$; and (c) social inhibition $\rho$ to maximize the reward rate (RR) in the indiscriminate stop-signalling model. The best tunings of parameters vary considerably with the recruitment being mostly low.

social inhibition (for example, figure 10a,b). If the system is not poised close to a bifurcation, the dynamics between switches roughly linearly decays to the stable equilibrium. However, in the discriminate stop-signalling model, the system can lie close to the saddle-node bifurcation beyond which the model exhibits bistability (figure 10c). In this case, the ghost of the saddle-node slows the solution trajectory, a nonlinear effect which is not well characterized by a linear approximation [70].

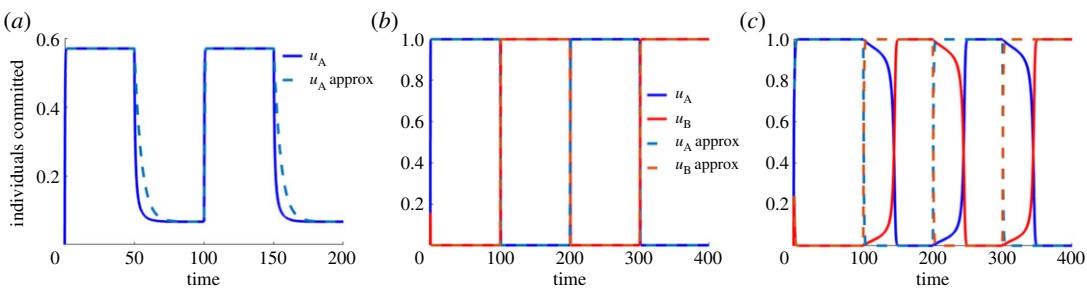

**Figure 10.** Linear approximation of the switch induced periodic solutions is generally accurate in (*a*) the single-feeder choice model ($\bar{\alpha} = 2$, $T = 50$ min, $\gamma = 2.8$ and $\beta = 3$) and (*b*) two-feeder choice model (direct switching here with model parameters $\bar{\alpha} = 2$, $T = 100$ min, $\gamma = 0.01$, $\beta = 0.1$ and $\rho = 10$). (*c*) However, when studying the discriminate stop-signalling model close to the saddle-node bifurcation, nonlinear effects shape the periodic solution of the full model equation (2.1) in ways that are not well approximated by the linearizations. Model parameters are $\bar{\alpha} = 2$, $T = 100$ min, $\gamma = 0.01$, $\beta = 3.6$ and $\rho = 1$.

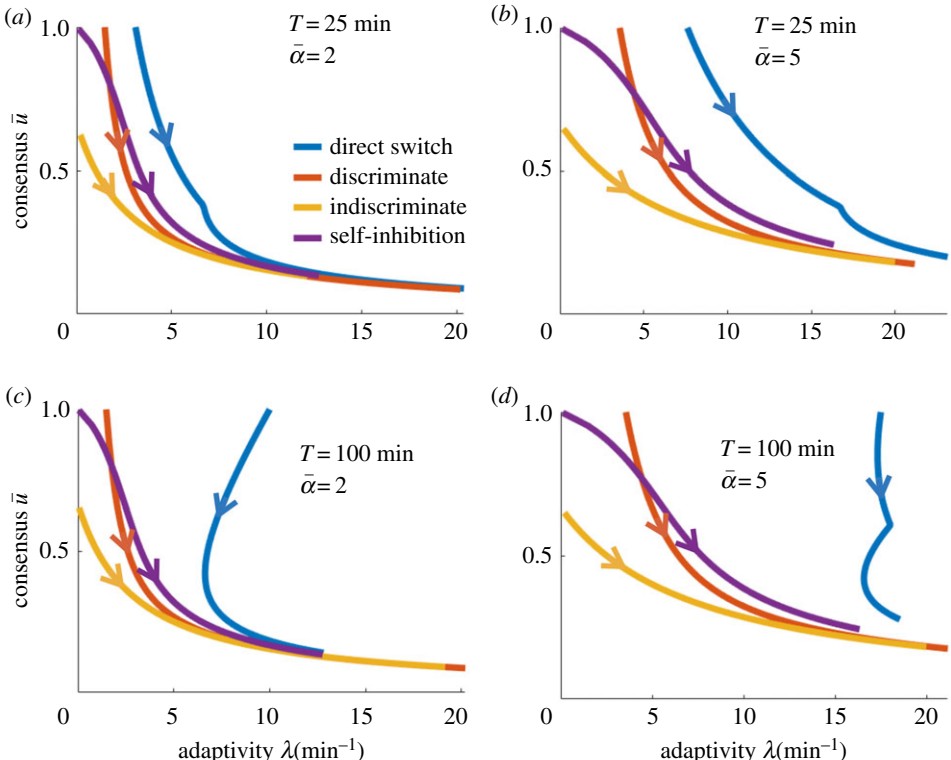

**Figure 11.** Consensus $\bar{u}$ and adaptivity $\lambda$ computed as described in §§B(b) and B(d), as abandonment rate $\gamma$ is increased between $[0, 20]$ min$^{-1}$ (along the direction of the arrows) for all models. Other parameters are fixed at their optimum level.

## (d) Computing adaptivity and consensus across models

Here, we calculate consensus $\bar{u}$ and adaptivity $\lambda$ across a wider range of environments as the abandonment rate $\gamma$ (figure 11) and social inhibition rate $\rho$ (figure 12) are varied. The general trends observed in figure 6*b,c* are preserved. For strong enough abandonment $\gamma$, adaptivity $\lambda$ increases as $\bar{u}$ decreases, and direct switching tends to balance this trade-off best (figure 11). Indiscriminate stop-signalling presents a similar trade-off as social inhibition strength $\rho$ is increased (figure 12), while the other social inhibition mechanisms eventually show increases in both consensus $\bar{u}$ and adaptivity $\lambda$, but again direct switching tends to provide higher levels of both overall.

## (e) Effect of heterogeneity in recruitment and abandonment

Here, we introduce and simulate a model of a colony whose bees have recruitment and forgetting rates drawn from a distribution $p(\beta, \gamma)$. Here, each parameter $\beta$ and $\gamma$ is drawn independently

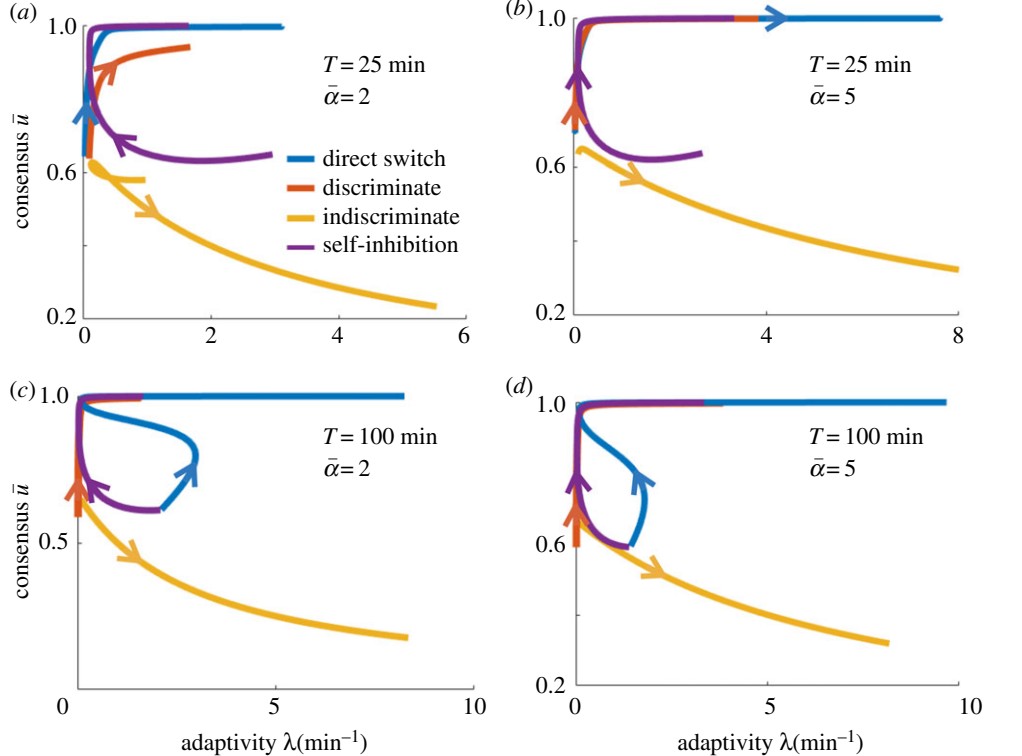

**Figure 12.** Consensus $\bar{u}$ and adaptivity $\lambda$ computed as described in §§B(b) and B(d), as social inhibition strength $\rho$ is increased between [0, 20] min$^{-1}$ (along the direction of the arrows) for all models. Other parameters are fixed at their optimal level.

from a gamma distribution $\Gamma(a, b)$ with shape $a$ and rate $b$ whose mean $a/b$ is set equal to the recruitment and forgetting rates of the mean-field equation (2.1). In this framework, $u_A$ and $u_B$ are probability density functions of $\beta$ and $\gamma$ evolving in $t$. Initially, all bees are in the uncommitted state $u_U(\beta, \gamma, 0) = p(\beta, \gamma)$ and $u_A(\beta, \gamma, 0) \equiv u_B(\beta, \gamma, 0) \equiv 0$ and the population fractions subsequently evolve

$$\frac{\partial u_A(\beta, \gamma, t)}{\partial t} = \left(\alpha_A + \int_0^\infty \int_0^\infty \beta u_A(\beta, \gamma, t)\right)[p(\beta, \gamma) - u_A(\beta, \gamma, t) - u_B(\beta, \gamma, t)] - \gamma u_A(\beta, \gamma, t)$$
$$- \rho u_A(\beta, \gamma, t) u_B(\beta, \gamma, t)(\alpha_B - \alpha_A) \tag{B 1a}$$

and

$$\frac{\partial u_B(\beta, \gamma, t)}{\partial t} = \left(\alpha_B + \int_0^\infty \int_0^\infty \beta u_B(\beta, \gamma, t)\right)[p(\beta, \gamma) - u_A(\beta, \gamma, t) - u_B(\beta, \gamma, t)] - \gamma u_B(\beta, \gamma, t)$$
$$- \rho u_A(\beta, \gamma, t) u_B(\beta, \gamma, t)(\alpha_A - \alpha_B), \tag{B 1b}$$

and note that the social inhibition term obeys direct switching. When setting the mean recruitment and abandonment rates to be the optimal ones identified in figure 4, introducing some heterogeneity decreases both consensus and adaptivity of the foraging colony slightly (figure 13a,c), and this effect grows for high-variance distributions (figure 13b,d).

## (f) Stochastic effects in a finite-sized system

Honeybee colonies tend to be of modest size (in the 1000s) [6], and so it is reasonable to expect some impact of finite-size effect on the dynamics of foraging. In general, we found finite-size effects induced fluctuations about the typical mean periodic switching solutions, but that it did not qualitatively alter the general behaviour of the foraging colony (figure 14). The finite-size model is governed by a master equation, determining the probability of all possible changes in committed and uncommitted populations.

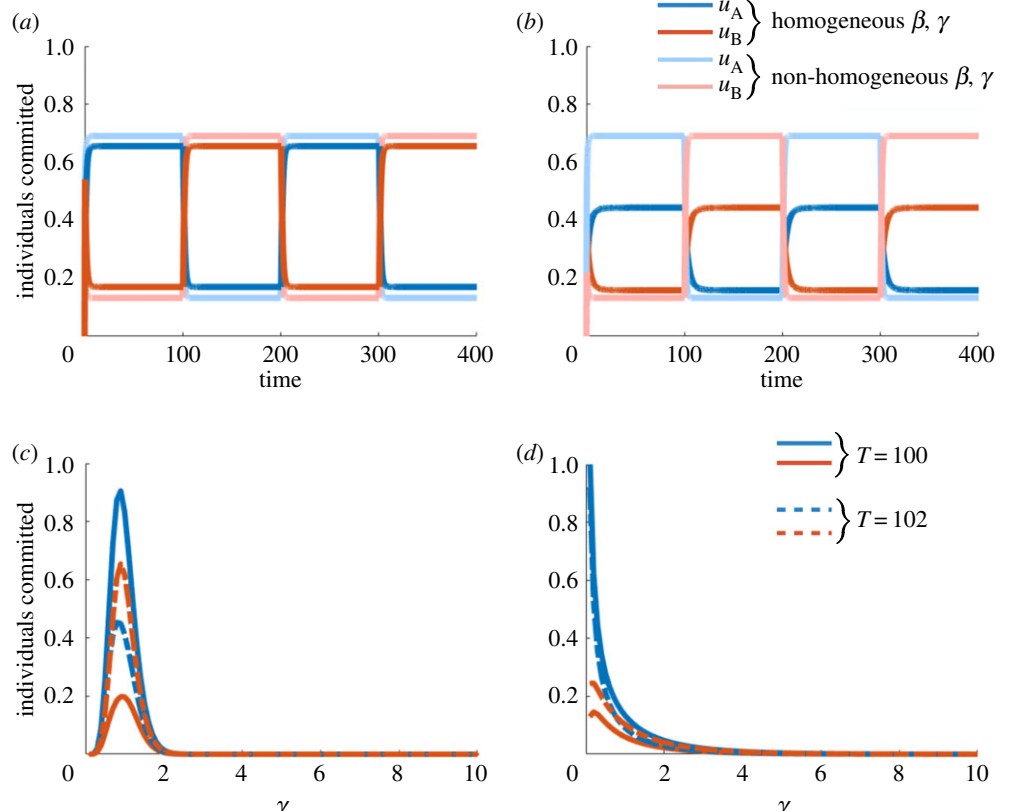

**Figure 13.** Heterogeneity in the recruitment and forgetting rates can decrease consensus and adaptivity. Consensus slightly decreases when simulating equation (B 1) with (*a*) weak heterogeneity $\beta \in \Gamma$ (20, 0.1) and $\gamma \in \Gamma$ (10, 0.1) (heavy lines) as opposed to (light lines) constant recruitment ($\beta = 2$) and forgetting ($\gamma = 1$) as in equation (2.1), and (*b*) consensus decreases more when more dramatic heterogeneity is considered: $\beta \in \Gamma$ (1, 2) and $\gamma \in \Gamma$ (0.5, 2). Adaptivity decreases with heterogeneity as well, as we see for (*c*) weak heterogeneity distribution, individuals still abandon relatively quickly after a change point, but (*d*) more slowly when rates are drawn from distribution with strong heterogeneity. Distributions are independent, so $p(\beta, \gamma) = p(\beta)p(\gamma)$; switching period $T = 100$ min; other parameters $\alpha = 2$, $\rho = 1$, and $\tau = 0$.

In the case of the single-feeder model, the master equation for the probability $p(n, t)$ of finding $n$ bees committed to foraging at time $t$ is

$$\dot{p}(n, t) = r_+(n - 1)p(n - 1, t) + r_-(n + 1)p(n + 1, t) - [r_+(n) + r_-(n)]p(n, t), \qquad (C\ 2)$$

for integer $n = 0, 1, 2, \ldots, N$ with boundary conditions $p(-1, t) = p(N + 1, t) = 0$ and forward and backward transition rates

$$r_+(n) = (N - n)(\tilde{\alpha}(t) + \tilde{\beta}n) \quad \text{and} \quad r_-(n) = \tilde{\gamma}n + \tilde{\rho}(\bar{\tilde{\alpha}} - \tilde{\alpha}(t - \tau))n^2$$

for system size (total bee number) $N$. To obtain the mean-field equation (2.3) as $N \to \infty$ [26,71], one must define $\tilde{\alpha}(t) = \alpha(t)/N$, $\tilde{\beta} = \beta/N^2$, $\tilde{\gamma} = \gamma/N$ and $\tilde{\rho} = \rho/N^2$. Note, the scalings correspond to the power of the population count appearing in the interaction term, ensuring the transition terms remain bounded in the thermodynamic limit. We used the stochastic simulation algorithm by Gillespie [72] to evolve the stochastic system for the statistic plotted in figure 14*a,b*. We make two remarks about our findings. First, the colony generally increases the fraction of committed foragers when food is present at the feeder and decreases when food is removed. Second, the amplitude of fluctuations in individual simulations decreases with system size, as typically expected [71], as evidenced by the narrower standard deviations in the solution trajectories in the $N = 1000$ versus the $N = 100$ simulations.

In the case of the two-feeder model, the master equation is more complicated as it must track the probability of transitions between uncommitted bees, bees committed to feeder A, and those committed to feeder B. Indeed, we can write the model down for any of the four forms of social

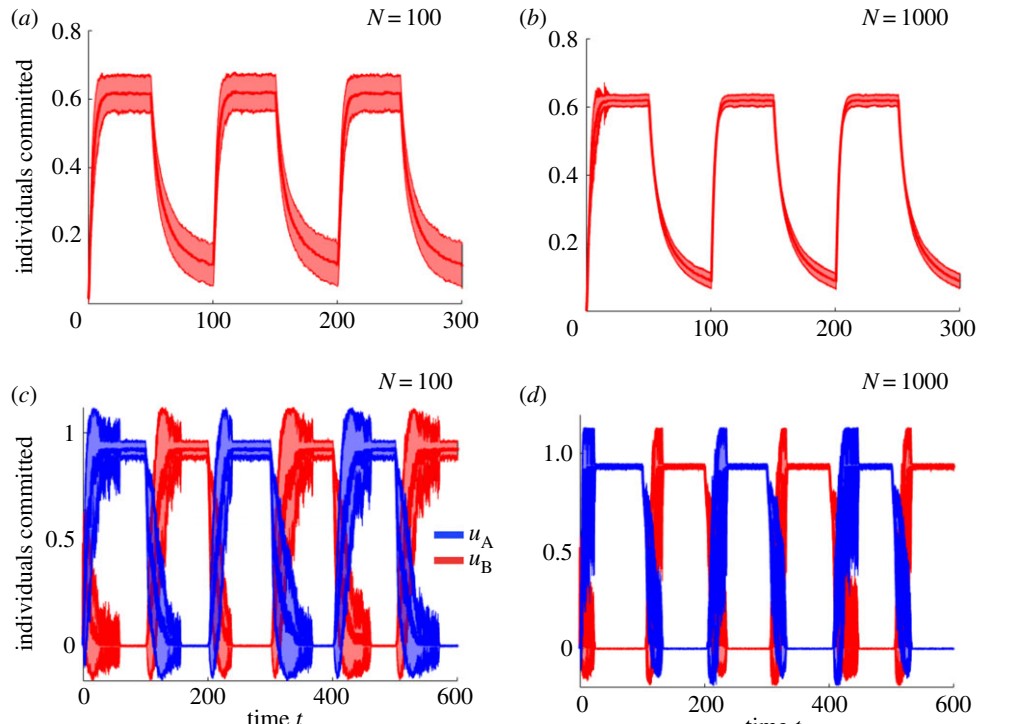

**Figure 14.** Mean and standard deviation of stochastic simulations of the single-feeder equation (B 2) and two-feeder equation (B 3) in the case of temporal switching of feeder quality occurring at $T = 50$ min intervals. In the single-feeder model, for (*a*) $N = 100$ and (*b*) $N = 1000$, near periodic switching of the mean trajectory of simulations (lines) is not far from the behaviour of the mean-field system equation (2.3). Other model parameters are $\tilde{\alpha} = 0.2$, $\tilde{\beta} = 0.2$ and $\tilde{\gamma} = 0.2$. Standard deviations (shaded regions) decrease as the system size is increased. Similar trends are apparent in the statistics of the two-feeder finite-sized models with discriminate stop signalling (*c*) $N = 100$ and (*d*) $N = 1000$. Other model parameters are $\tilde{\alpha} = 2$, $\tilde{\beta} = 2$, $\tilde{\gamma} = 0.1$, $\tilde{\rho} = 1$, $T = 100$ min and $\tau = 0$.

inhibition, but we just provide the discriminate stop-signalling model here. Others can be written similarly. The probability $p(n_A, n_B, t)$ of finding $n_A$ bees committed to A and $n_B$ committed to $B$ at time $t$ given system size $N$ is (dropping the argument in $t$ for brevity):

$$\dot{p}(n_A, n_B) = r_{0A}(n_A - 1, n_B)p(n_A - 1, n_B) + r_{0B}(n_A, n_B - 1) + r_{A0}(n_A + 1, n_B)$$
$$+ r_{B0}(n_A, n_B + 1)p(n_A, n_B + 1) - [r_{0A}(n_A, n_B) + r_{0B}(n_A, n_B)$$
$$+ r_{A0}(n_A, n_B) + r_{B0}(n_A, n_B)]p(n_A, n_B), \tag{C 3}$$

for $n_A, n_B = 0, 1, \ldots, N$ with the condition that $n_A + n_B \leq N$, boundary conditions $p(-1, n_B) = p(n_A, -1) = p(N + 1, n_B) = p(n_A, N + 1) = 0$, and transition rates

$$r_{0A}(n_A, n_B) = (N - n_A - n_B)(\tilde{\alpha}_A(t) + \tilde{\beta}n_A), \quad r_{0B}(n_A, n_B) = (N - n_A - n_B)(\tilde{\alpha_A}(t) + \tilde{\beta}n_B)$$

and

$$r_{A0}(n_A, n_B) = \tilde{\gamma}n_A + \tilde{\rho}\alpha_B(t - \tau)n_A n_B, \quad r_{B0}(n_A, n_B) = \tilde{\gamma}n_B + \tilde{\rho}\alpha_A(t - \tau)n_A n_B.$$

As in the single-feeder model, periodic switching with environmental switches is apparent, and the amplitude of fluctuations decreases with system size (figure 14*c,d*).

A detailed study of the finite-size population model would require a much more thorough treatment and statistical analysis. We expect the effects of stochasticity will not considerably impact our general findings. The only qualitative differences we would expect would be in the case of unrealistically small systems (e.g. $N = 10$), and in bistable systems (like cases of the discriminate stop-signalling model), where fluctuations could drive switching between multiple stable equilibria [66].

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
