## [Reviewer comments · Royal Society Open Science]

Review History

RSOS-191277.R0 (Original submission)

Review form: Reviewer 1

Is the manuscript scientifically sound in its present form?

Yes

Are the interpretations and conclusions justified by the results?

Yes

Is the language acceptable?

Yes

Do you have any ethical concerns with this paper?

No

Have you any concerns about statistical analyses in this paper?

No

Recommendation?

Accept with minor revision (please list in comments)

Comments to the Author(s)

This manuscript develops a mathematical model of honeybee swarms to study the dynamics of group foraging from feeders with temporally switching food quality. It shows that social interactions improve foraging from a single feeder if temporal switching is fast or feeder quality is low. Given multiple feeders, the most effective form of social interaction is direct switching.

Overall, the manuscript is well written and scientifically sound. The mathematical analysis and numerical simulation are good to support the conclusions. I only have a few comments:

(1) The authors may wish to discuss the difference of this manuscript to their previous works such as ref. [9], [26].

(2) If I understand correctly, the parameters β (i.e. rate at which bees recruit nest mates to feeder via waggle dancing) and γ (i.e. rate at which bees spontaneously abandon a feeder) are constant for all bees. Is it possible to explore the effect of heterogeneity (e.g. gamma distribution) in these parameters?

Review form: Reviewer 2

Is the manuscript scientifically sound in its present form?

No

Are the interpretations and conclusions justified by the results?

No

Is the language acceptable?

Yes

Do you have any ethical concerns with this paper?

No

Have you any concerns about statistical analyses in this paper?

No

Recommendation?

Reject

Comments to the Author(s)

Dear authors and editors,

I am a weird conflict regarding the manuscript. On the one hand I can not judge the mathematical modeling, on the other hand I do not think that the authors have clearly understood foraging behavior and foraging recruitment in honey bees.

For example, the authors use the term swarm during the whole manuscript, although they mean colony. In honey bees the swarm is a temporary phase during which the colony searches for a

new nest site, swarms do not really forage. Second, the distinction between scouts and recruits is thought to be more robust and an individually does not spontaneously shift between two behaviors. Third in my opinion foraging groups in honey bees do not necessarily stop foraging at a low value feeder.

From my perspective the modeling might good, but the biology is quite wrong. So, then the questions arises what is the modeling good for.

Decision letter (RSOS-191277.R0)

28-Aug-2019

Dear Professor Kilpatrick:

Manuscript ID RSOS-191277 entitled "Social inhibition maintains adaptivity and consensus of foraging honeybee swarms in dynamic environments" which you submitted to Royal Society Open Science, has been reviewed. The comments from reviewers are included at the bottom of this letter.

In view of the criticisms of the reviewers, the manuscript has been rejected in its current form. However, a new manuscript may be submitted which takes into consideration these comments.

Please note that resubmitting your manuscript does not guarantee eventual acceptance, and that your resubmission will be subject to peer review before a decision is made.

Your resubmitted manuscript should be submitted by 25-Feb-2020. If you are unable to submit by this date please contact the Editorial Office.

on behalf of Professor Matjaz Perc (Associate Editor) and Mark Chaplain (Subject Editor)
openscience@royalsociety.org

Reviewers' Comments to Author:

Reviewer: 1

Comments to the Author(s)

This manuscript develops a mathematical model of honeybee swarms to study the dynamics of group foraging from feeders with temporally switching food quality. It shows that social interactions improve foraging from a single feeder if temporal switching is fast or feeder quality is low. Given multiple feeders, the most effective form of social interaction is direct switching.

Overall, the manuscript is well written and scientifically sound. The mathematical analysis and numerical simulation are good to support the conclusions. I only have a few comments:

(1) The authors may wish to discuss the difference of this manuscript to their previous works such as ref. [9], [26].

(2) If I understand correctly, the parameters β (i.e. rate at which bees recruit nest mates to feeder via waggle dancing) and γ (i.e. rate at which bees spontaneously abandon a feeder) are constant for all bees. Is it possible to explore the effect of heterogeneity (e.g. gamma distribution) in these parameters?

Reviewer: 2

Comments to the Author(s)

Dear authors and editors,

I am a weird conflict regarding the manuscript. On the one hand I can not judge the mathematical modeling, on the other hand I do not think that the authors have clearly understood foraging behavior and foraging recruitment in honey bees.

For example, the authors use the term swarm during the whole manuscript, although they mean colony. In honey bees the swarm is a temporary phase during which the colony searches for a new nest site, swarms do not really forage. Second, the distinction between scouts and recruits is thought to be more robust and an individual does not spontaneously shift between two behaviors. Third in my opinion foraging groups in honey bees do not necessarily stop foraging at a low value feeder.

From my perspective the modeling might be good, but the biology is quite wrong. So, then the question arises what is the modeling good for.

Author's Response to Decision Letter for (RSOS-191277.R0)

See Appendix A.

RSOS-191681.R0

Review form: Reviewer 1

Is the manuscript scientifically sound in its present form?

Yes

Are the interpretations and conclusions justified by the results?

Yes

Is the language acceptable?

Yes

Do you have any ethical concerns with this paper?

No

Have you any concerns about statistical analyses in this paper?

No

Recommendation?

Accept as is

Comments to the Author(s)

The authors have fully addressed my comments. My pleasure to recommend its acceptance for the Royal Society Open Science. Congratulations!

Review form: Reviewer 2

Is the manuscript scientifically sound in its present form?

Yes

Are the interpretations and conclusions justified by the results?

Yes

Is the language acceptable?

Yes

Do you have any ethical concerns with this paper?

No

Have you any concerns about statistical analyses in this paper?

No

Recommendation?

Accept as is

Comments to the Author(s)

Dear authors,

Thanks you very much for your detailed and appropriate responses to my concerns. One minor thing, you might check the following paper and cite it if you agree that it is appropriate for your manuscript.

Townsend-Mehler, J. M., & Dyer, F. C. (2011). An integrated look at decision-making in bees as they abandon a depleted food source. *Behavioral Ecology and Sociobiology*, 66(2), 275–286.

Review form: Reviewer 3

Is the manuscript scientifically sound in its present form?

Yes

Are the interpretations and conclusions justified by the results?

Yes

Is the language acceptable?

Yes

Do you have any ethical concerns with this paper?

No

Have you any concerns about statistical analyses in this paper?

No

Recommendation?

Major revision is needed (please make suggestions in comments)

Comments to the Author(s)

First, I do not believe I had reviewed this manuscript before; reviewer 1's comments do not look familiar or what I would usually write, and I cannot find a previous invitation for the first submission. So I think I am looking at this with fresh eyes, and have some new comments.

Second, I disagree with several of reviewer 2's criticisms (notwithstanding correct usage of terminology, which I agree with). The authors' responses are sound in my opinion. In particular if recruits and scouts are distinct castes then there is no scope for positive feedback to good quality nest / forage sites, since recruits will not recruit, and scouts will not follow. Hence models over the previous several decades are typically considering these as distinct behavioural states, as the authors respond. These models usually implicitly or explicitly consider only the forager class of bees (a small fraction of the total).

I like this work but have a general criticism around the objective function assumed; reward rate is probably fine as a fast-timescale criterion, but I think the lack of crowding in the reward rate function needs to be addressed. As it stands, the simple linear dependence of intake on proportion of scouts is intuitive and straightforward to see, apart from the need to adapt to changes in forage patch quality. Yet the authors repeatedly refer to crowding in discussing the natural history. Foraging models with crowding can yield much more complex optimal strategies, even with static options; see, for example, Talamali et al.

(<https://doi.org/10.1007/s11721-019-00176-9> - Fig 10). I can see two solutions for the authors: (1) do a lot more work including redoing the optimality analysis, (2) acknowledge that crowding is neglected and motivate this, e.g. by a very-large-forage patch assumption.

I think neglected some relevant results in one of the references they cite. Pais et al. [10] analysed the dynamics of their stop-signal model in a periodically changing environment - they showed a hysteresis loop for this model.

Regarding the result that indiscriminate stop-signalling does not admit bistability, it is worth noting that a similar analysis was done for the model of Seeley et al. (2012) in the supplementary material for that paper.

Minor comments:

Introduction

p.1, l.50: I don't think Seeley et al. interpreted the stop-signalling as dissuading recipients of signals from commitment to 'less suitable nest sites', since signalling was assumed to be targeted but symmetric; thus it was simply targeting individuals with different information.

p.2, l.9: I don't think nest sites change at a relevant timescale for decision-making... site qualities will change between swarming events.

footnote 3: I don't think consideration of evidence-weighting in collectives is that novel; abandonment implements leak from recruiter populations which is well known as implementing a weighted average for neural accumulators in neuroscience, and considered similarly by some social insect modellers.

Discussion

p.11, l.43: I think ref [29] should also be added to the references here, as the form of models presented there inspired several subsequent studies.

Decision letter (RSOS-191681.R0)

04-Nov-2019

Dear Professor Kilpatrick

On behalf of the Editor, I am pleased to inform you that your Manuscript RSOS-191681 entitled "Social inhibition maintains adaptivity and consensus of honey bees foraging in dynamic environments" has been accepted for publication in Royal Society Open Science subject to minor revision in accordance with the referee suggestions. Please find the referees' comments at the end of this email.

The reviewers and Subject Editor have recommended publication, but also suggest some minor revisions to your manuscript. Therefore, I invite you to respond to the comments and revise your manuscript.

- Ethics statement

- Data accessibility

<http://datadryad.org/submit?journalID=RSOS&manu=RSOS-191681>

- Competing interests

- Authors' contributions

- Acknowledgements

- Funding statement

Because the schedule for publication is very tight, it is a condition of publication that you submit the revised version of your manuscript before 13-Nov-2019. Please note that the revision deadline will expire at 00.00am on this date. If you do not think you will be able to meet this date please let me know immediately.

on behalf of Professor Matjaz Perc (Associate Editor) and Mark Chaplain (Subject Editor)
openscience@royalsociety.org

Reviewer comments to Author:

Reviewer: 1

Comments to the Author(s)

The authors have fully addressed my comments. My pleasure to recommend its acceptance for the Royal Society Open Science. Congratulations!

Reviewer: 3

Comments to the Author(s)

First, I do not believe I had reviewed this manuscript before; reviewer 1's comments do not look familiar or what I would usually write, and I cannot find a previous invitation for the first submission. So I think I am looking at this with fresh eyes, and have some new comments.

Second, I disagree with several of reviewer 2's criticisms (notwithstanding correct usage of terminology, which I agree with). The authors' responses are sound in my opinion. In particular if recruits and scouts are distinct castes then there is no scope for positive feedback to good quality nest / forage sites, since recruits will not recruit, and scouts will not follow. Hence models over the previous several decades are typically considering these as distinct behavioural states, as the authors respond. These models usually implicitly or explicitly consider only the forager class of bees (a small fraction of the total).

I like this work but have a general criticism around the objective function assumed; reward rate is probably fine as a fast-timescale criterion, but I think the lack of crowding in the reward rate function needs to be addressed. As it stands, the simple linear dependence of intake on proportion of scouts is intuitive and straightforward to see, apart from the need to adapt to changes in forage patch quality. Yet the authors repeatedly refer to crowding in discussing the natural history. Foraging models with crowding can yield much more complex optimal strategies, even with static options; see, for example, Talamali et al.

(<https://doi.org/10.1007/s11721-019-00176-9> - Fig 10). I can see two solutions for the authors: (1) do a lot more work including redoing the optimality analysis, (2) acknowledge that crowding is neglected and motivate this, e.g. by a very-large-forage patch assumption.

I think neglected some relevant results in one of the references they cite. Pais et al. [10] analysed the dynamics of their stop-signal model in a periodically changing environment - they showed a hysteresis loop for this model.

Regarding the result that indiscriminate stop-signalling does not admit bistability, it is worth noting that a similar analysis was done for the model of Seeley et al. (2012) in the supplementary material for that paper.

Minor comments:

Introduction

p.1, l.50: I don't think Seeley et al. interpreted the stop-signalling as dissuading recipients of signals from commitment to 'less suitable nest sites', since signalling was assumed to be targeted but symmetric; thus it was simply targeting individuals with different information.

p.2, l.9: I don't think nest sites change at a relevant timescale for decision-making... site qualities will change between swarming events.

footnote 3: I don't think consideration of evidence-weighting in collectives is that novel; abandonment implements leak from recruiter populations which is well known as implementing a weighted average for neural accumulators in neuroscience, and considered similarly by some social insect modellers.

Discussion

p.11, l.43: I think ref [29] should also be added to the references here, as the form of models presented there inspired several subsequent studies.

Reviewer: 2

Comments to the Author(s)

Dear authors,

thanks you very much for your detailed and appropriate responses to my concerns. One minor thing, you might check the following paper and cite it if you agree that it is appropriate for your manuscript.

Townsend-Mehler, J. M., & Dyer, F. C. (2011). An integrated look at decision-making in bees as they abandon a depleted food source. *Behavioral Ecology and Sociobiology*, 66(2), 275–286.

Author's Response to Decision Letter for (RSOS-191681.R0)

See Appendix B.

Decision letter (RSOS-191681.R1)

13-Nov-2019

Dear Professor Kilpatrick,

It is a pleasure to accept your manuscript entitled "Social inhibition maintains adaptivity and consensus of honey bees foraging in dynamic environments" in its current form for publication in Royal Society Open Science. The comments of the reviewer(s) who reviewed your manuscript are included at the foot of this letter.

Due to rapid publication and an extremely tight schedule, if comments are not received, your paper may experience a delay in publication. Royal Society Open Science operates under a

continuous publication model. Your article will be published straight into the next open issue and this will be the final version of the paper. As such, it can be cited immediately by other researchers. As the issue version of your paper will be the only version to be published I would advise you to check your proofs thoroughly as changes cannot be made once the paper is published.

on behalf of Professor Matjaz Perc (Associate Editor) and Mark Chaplain (Subject Editor)
openscience@royalsociety.org

Appendix A

September 24, 2019

Royal Society Open Science
Editorial Board

Dear Editors,

We thank the referees for their feedback on our manuscript, now titled “Social inhibition maintains adaptivity and consensus of honey bees foraging in dynamic environments.” Their suggestions were helpful in identifying ways to demonstrate the generalizability of our model, and to link better with existing biological and modeling literature. A detailed description of our changes are given below, and we have highlighted changes in the manuscript in blue.

We hope our manuscript is now suitable for publication in Royal Society Open Science and look forward to hearing from you.

Zachary P. Kilpatrick, PhD
Assistant Professor
Department of Applied Mathematics
University of Colorado Boulder
email: zpkilpat@colorado.edu

Reviewers' & editor's comments are italicized. Our responses are in plain text. Changes to the manuscript text are in blue.

Referee 1

Reviewer 1: This manuscript develops a mathematical model of honeybee swarms to study the dynamics of group foraging from feeders with temporally switching food quality. It shows that social interactions improve foraging from a single feeder if temporal switching is fast or feeder quality is low. Given multiple feeders, the most effective form of social interaction is direct switching. Overall, the manuscript is well written and scientifically sound. The mathematical analysis and numerical simulation are good to support the conclusions.

Thank you for your positive evaluation of our work.

I only have a few comments:

(1) The authors may wish to discuss the difference of this manuscript to their previous works such as ref. [9], [26].

We have now added a few sentences to the Discussion to clarify the difference between our work and that of Marshall et al (2009) and Seeley et al (2012):

Previous computational modeling studies of honeybee collective decisions primarily focused on groups solving house-hunting problems in static environments [13,20], emphasizing how social interactions shape the speed at which consensus is obtained within a collective. However, less attention has been paid to how such collectives must adapt to change, and how social communication determines group adaptivity. Some previous work has discussed the importance of uncommitted

Figure 1: Heterogeneity in the recruitment and forgetting rates can decrease consensus and adaptivity. Consensus slightly decreases when simulating Eq. (19) with (a) weak heterogeneity $\beta \in \Gamma(20, 0.1)$ and $\gamma \in \Gamma(10, 0.1)$ (heavy lines) as opposed to (light lines) constant recruitment ($\beta = 2$) and forgetting ($\gamma = 1$) as in Eq. (1), and (b) consensus decrease more when more dramatic heterogeneity is considered: $\beta \in \Gamma(1, 2)$ and $\gamma \in \Gamma(0.5, 2)$. Adaptivity decreases with heterogeneity as well, as we see for (c) weak heterogeneity distribution individuals still abandon relatively quickly after a changepoint, but (d) more slowly when rates are drawn from distribution with strong heterogeneity. Distributions are independent, so $p(\beta, \gamma) = p(\beta)p(\gamma)$; switching period $T = 100\text{min}$; other parameters $\alpha = 2$, $\rho = 1$, and $\tau = 0$.

inspector bees in affording group adaptivity [8], but our work is the first to systematically compare how different forms of social communication [3, 13, 15, 20] shape group adaptivity. Social communication by which one bee can switch the foraging preference of another appear to be incredibly effective in providing groups with the ability to both build consensus and adaptive to change.

If I understand correctly, the parameters β (i.e. rate at which bees recruit nest mates to feeder via waggle dancing) and γ (i.e. rate at which bees spontaneously abandon a feeder) are constant for all bees. Is it possible to explore the effect of heterogeneity (e.g. gamma distribution) in these parameters?

Indeed, in general, we had expected our qualitative results would remain unchanged under consideration of such heterogeneity. We have now explored this concretely by developing and simulating a model in which β and γ are drawn from distributions. As explained in our added text, appendix, and figure, our results were fairly robust to the inclusion of such heterogeneity.

Our findings are fairly robust to considerations of interaction heterogeneity within the colony (See Appendix C.6 and Fig. 14). A colony whose bees have individualized rates of recruitment and abandonment exhibited slight decreases in consensus and adaptivity, but qualitatively the group still remained responsive to change.

Appendix C.5: Effect of heterogeneity in recruitment and forgetting. Here we introduce and simulate a model of a

colony whose bees have recruitment and forgetting rates drawn from a distribution $p(\beta, \gamma)$. Here each parameter β and γ is drawn independently from a gamma distribution $\Gamma(a, b)$ with shape a and rate b whose mean a/b is set equal to the recruitment and forgetting rates of the mean field Eq. (1). In this framework u_A and u_B are probability density functions of β and γ evolving in t . Initially, all bees are in the uncommitted state $u_U(\beta, \gamma, 0) = p(\beta, \gamma)$ and $u_A(\beta, \gamma, 0) \equiv u_B(\beta, \gamma, 0) \equiv 0$ and the population fractions subsequently evolve

$$\frac{\partial u_A(\beta, \gamma, t)}{\partial t} = \left(\alpha_A + \int_0^\infty \int_0^\infty \beta u_A(\beta, \gamma, t) \right) [p(\beta, \gamma) - u_A(\beta, \gamma, t) - u_B(\beta, \gamma, t)] - \gamma u_A(\beta, \gamma, t) \quad (1a)$$

$$- \rho u_A(\beta, \gamma, t) u_B(\beta, \gamma, t) (\alpha_B - \alpha_A)$$

$$\frac{\partial u_B(\beta, \gamma, t)}{\partial t} = \left(\alpha_B + \int_0^\infty \int_0^\infty \beta u_B(\beta, \gamma, t) \right) [p(\beta, \gamma) - u_A(\beta, \gamma, t) - u_B(\beta, \gamma, t)] - \gamma u_B(\beta, \gamma, t) \quad (1b)$$

$$- \rho u_A(\beta, \gamma, t) u_B(\beta, \gamma, t) (\alpha_A - \alpha_B),$$

and note that the social inhibition term obeys direct switching. When setting the mean recruitment and abandonment rates to be the optimal ones identified in Fig. 4, introducing some heterogeneity decreases both consensus and adaptivity of the foraging colony slightly (Fig. 13a,c), and this effect grows for high variance distributions (Fig. 13b,d).

Reviewer 2: I am a weird conflict regarding the manuscript. On the one hand I can not judge the mathematical modeling, on the other hand I do not think that the authors have clearly understood foraging behavior and foraging recruitment in honey bees.

The goal of this work was to provide a minimal model of social interactions within honeybee colonies that accounts for their ability to adapt to environmental change, and to determine how social interactions shape foraging profitability. Certainly, we could have written down a much more complicated mathematical model to begin with, but it would have been less amenable to our analysis and insights, and we expect the core conclusions would be qualitatively similar.

Through an extensive look at the literature, we found that worker bees can transition between multiple types of behavior (e.g., scouting, recruiting) depending on changing environmental demands [2, 18]. Our focus is solely on the foraging process, and so our model incorporates the dynamics of field bees but not that of house bees. We have now updated the paragraph preceding our introduction of the model to more accurately reflect this.

Note also that we assume the number of foraging bees is large enough (over 10,000) as to make a mean field approximation reasonable, consistent with field counts of bees [22]. As a result, even if individual bees did not transition between being scouts and recruits, the variables u_A and u_B could then be conceived of as the sum of scouts and recruits foraging at site A and B , so that a fraction of each population would engage in each type of social interaction. This would still be captured by our model by conceiving of the transition rates α , β , γ , and ρ as lumping together these fractions with the propensity to commit, recruit, abandon, or socially inhibit within that particular class of bee. This is now discussed in the paragraph we have added to the Discussion justifying our model as one where bees have flexible roles, which we show below in our response to your scouts/recruits comment.

For example, the authors use the term swarm during the whole manuscript, although they mean colony. In honey bees the swarm is a temporary phase during which the colony searches for a new nest site, swarms do not really forage.

Thank you for pointing this out. We have modified the terminology throughout to more accurately reflect the state of the group of honeybees in our model, removing use of the word ‘swarm’ when inappropriate.

We have also added a paragraph to the Introduction clarifying the distinction between studying house-hunting versus foraging, and how social interactions could differentially affect the colony’s goals in each case:

To study how social inhibition shapes foraging yields, we focus on a task in which the nectar quality of feeders is switched periodically. Related situations likely occur in nature due to the dynamics of competitor and predator prevalence, crowding

by nestmates, and weather fluctuations [5, 12, 14]. Temporally periodic changes may not occur naturally but can be easily generated in controlled experiments [8, 23]. There are important distinctions between the goals of colonies in foraging as opposed to those searching for a new home site. Once a colony establishes a permanent nest site, this is the starting and ending point for each food foraging excursion. The colony need not reach consensus to obtain nutrition from foraging, since food is brought to the nest regardless of how many foraging sites the group is split between [12]. In contrast, when a honey bee swarm looks for a nest, it must reach consensus for all bees and the queen to fly to the selected site. If not, their transition to a permanent nest site will be delayed, or the swarm might split. Bees use stop-signals to obtain this needed consensus when house-hunting, especially when two potential sites are of similar quality [20]. Consensus is not essential when foraging for food, but as we will show increasing the fraction of the colony at the best foraging site increases foraging yields.

We return to the point in the Discussion with the following added paragraph now:

Most work studying the effects of social inhibition on honeybee colony decisions focuses on swarms choosing a place to build a nest from sites whose qualities are fixed in time [20]. The need for social inhibition is clear in this context since it promotes consensus, generating a consistent opinion across the swarm and preventing the deadlock and group splitting. On the other hand, it is not immediately obvious that social inhibition would improve foraging if the primary effect is to increase consensus, since colonies can obtain and store food even when foragers are split between multiple feeders, though stop-signals can reduce crowding [12]. Nonetheless, we found that when the colony can rapidly switch opinion so nearly all bees agree to forage from the most profitable feeder, this does increase the nutrition yield of the colony overall. However, consensus is only advantageous in dynamic environments if it does not come at a cost to adaptivity: the opinion around which consensus is built should change with the environment.

Second, the distinction between scouts and recruits is thought to be more robust and an individually does not spontaneously shift between two behaviors.

We understand that one view of scouting vs. recruiting behavior in honeybee collectives portrays individuals as fixed in their caste. However, there has also been extensive work challenging this notion, and demonstrating that bees will often perform tasks outside their normal repertoire when it is advantageous to the colony for them to do so [18]. For instance, non-foraging bees may begin foraging when the colony has a limited number of foragers [7], or foragers may switch to brood care when needed [16]. Moreover, it has even been suggested that a strict conceptualization dividing foraging bees into scouts and recruits is unrealistic [2]. Rather, each forager can be conceived of as having several different possible behavioral states they may switch between depending on the demands of the colony. We have now added a paragraph to the Discussion section to highlight this point:

Another possible extension of our model would be to consider separate populations of scouts and foraging recruits as in some previous modeling studies [4, 24]. Our analysis assumes bees can fluidly transition between scouting and foraging behavior, as documented in several previous studies [2, 18]. Overall, a strict and unchanging division of labor within the hive provides an incomplete description of colony organization. For instance, bees may switch to foraging when the environment demands it [7] or when socially signaled to do so [9], and thus a strict caste divide between scouts and recruits may be unrealistic [2]. honey bees' roles appear to be strongly determined by the changing requirements of the colony, such as the influx or availability of nectar, rather than strictly due to some genetic predisposition [17, 18]. Bees that scout and forage tend to be in the same life cycle phase, and as such are more amenable to temporal caste switching [10]. Such flexibility may even be a rule rather than exception to colony labor organization [11].

Third in my opinion foraging groups in honey bees do not necessarily stop foraging at a low value feeder.

While we do value the reviewer's opinion, it would have been helpful if they could have pointed to a reference to support this opinion. We constructed our modeling study based on experimental observations reported in several previous papers. In early work, von Frisch (1967, 1971) showed bees tend to reduce the frequency of recruitment dances for feeders with lower concentration sugar solution than other higher available feeders [26, 27]. Bees appear to not prefer to collect such solutions for storage in the hive as converting highly dilute sugar solution into honey is both time and labor intensive.

Moreover, several authors have run experiments in which the quality of a feeder is switched in time. Seeley et al (1991) found a colony-level response to changes in feeder quality, whereby suddenly poor quality feeders are abandoned in favor of high quality feeders [23]. Granovski et al (2012) found something similar, highlighting the importance of scouting and inspection mechanisms in facilitating colony-wide adaptation. In addition, the decision of foragers as to whether they will perform recruitment dances or not are related to the quality of the sugar solution they have collected [19, 21]. Note, we had already summarized these points in the Introduction of the initial version of this manuscript.

“Bees adapt to change by abandoning less profitable nectar sources for those with higher yields [23], and by modifying the number of foragers [1, 25]. Prior studies focused on how waggle dance recruitment or the heterogeneity of individual bee roles shape colony adaptivity [6, 8].”

From my perspective the modeling might good, but the biology is quite wrong. So, then the questions arises what is the modeling good for.

Our model was constructed so that we could focus on the effects of different social inhibition mechanisms on the adaptivity and consensus features of colonies foraging in dynamic environments. There is clear evidence from previous work that bees will abandon less profitable feeders in favor of more profitable ones [8, 23], but the group-wide mechanism facilitating this is unresolved. Our contribution is to demonstrate that inhibitory mechanism that directly switch foragers’ preferences are most advantageous, but that other mechanisms like stop-signaling are efficacious too. This highlights the importance of distinguishing social inhibitory mechanisms in the context of foraging, and makes a testable prediction that colonies in more volatile environments may place a premium on utilizing mechanisms that promote adaptation.

References

- [1] Y. BEN-SHAHAR, A. ROBICHON, M. SOKOLOWSKI, AND G. ROBINSON, *Influence of gene action across different time scales on behavior*, *Science*, 296 (2002), pp. 741–744.
- [2] J. C. BIESMEIJER AND H. DE VRIES, *Exploration and exploitation of food sources by social insect colonies: a revision of the scout-recruit concept*, *Behavioral Ecology and Sociobiology*, 49 (2001), pp. 89–99.
- [3] N. BRITTON, N. FRANKS, S. PRATT, AND T. SEELEY, *Deciding on a new home: how do honeybees agree?*, *Proceedings of the Royal Society of London. Series B: Biological Sciences*, 269 (2002), pp. 1383–1388.
- [4] S. CAMAZINE, J. SNEYD, M. J. JENKINS, AND J. MURRAY, *A mathematical model of self-organized pattern formation on the combs of honeybee colonies*, *Journal of Theoretical Biology*, 147 (1990), pp. 553–571.
- [5] P. D. COOPER, W. M. SCHAFFER, AND S. L. BUCHMANN, *Temperature regulation of honey bees (*apis mellifera*) foraging in the sonoran desert*, *Journal of Experimental Biology*, 114 (1985), pp. 1–15.
- [6] A. DORNHAUS AND L. CHITTKA, *Why do honey bees dance?*, *Behavioral Ecology and Sociobiology*, 55 (2004), pp. 395–401.
- [7] J. FREE, *Managing honeybee colonies to enhance the pollen-gathering stimulus from brood pheromones*, *Applied Animal Ethology*, 5 (1979), pp. 173–178.
- [8] B. GRANOVSKIY, T. LATTY, M. DUNCAN, D. J. SUMPTER, AND M. BEEKMAN, *How dancing honey bees keep track of changes: the role of inspector bees*, *Behavioral Ecology*, 23 (2012), pp. 588–596.
- [9] Z.-Y. HUANG AND G. E. ROBINSON, *Social control of division of labor in honey bee colonies*, in *Information processing in social insects*, Springer, 1999, pp. 165–186.

- [10] B. R. JOHNSON, *Organization of work in the honeybee: a compromise between division of labour and behavioural flexibility*, Proceedings of the Royal Society of London. Series B: Biological Sciences, 270 (2003), pp. 147–152.
- [11] ———, *Division of labor in honeybees: form, function, and proximate mechanisms*, Behavioral ecology and sociobiology, 64 (2010), pp. 305–316.
- [12] C. W. LAU AND J. C. NIEH, *Honey bee stop-signal production: temporal distribution and effect of feeder crowding*, Apidologie, 41 (2010), pp. 87–95.
- [13] J. A. MARSHALL, R. BOGACZ, A. DORNHAUS, R. PLANQUÉ, T. KOVACS, AND N. R. FRANKS, *On optimal decision-making in brains and social insect colonies*, Journal of the Royal Society Interface, 6 (2009), pp. 1065–1074.
- [14] J. C. NIEH, *The stop signal of honey bees: reconsidering its message*, Behavioral Ecology and Sociobiology, 33 (1993), pp. 51–56.
- [15] ———, *A negative feedback signal that is triggered by peril curbs honey bee recruitment*, Current Biology, 20 (2010), pp. 310–315.
- [16] R. E. PAGE JR, G. E. ROBINSON, D. S. BRITTON, AND M. K. FONDRK, *Genotypic variability for rates of behavioral development in worker honeybees (*apis mellifera* l.)*, Behavioral Ecology, 3 (1992), pp. 173–180.
- [17] C. RIBBANDS, *Division of labour in the honeybee community*, Proceedings of the Royal Society of London. Series B-Biological Sciences, 140 (1952), pp. 32–43.
- [18] G. E. ROBINSON, *Regulation of division of labor in insect societies*, Annual review of entomology, 37 (1992), pp. 637–665.
- [19] R. SCHEINER, R. E. PAGE, AND J. ERBER, *Sucrose responsiveness and behavioral plasticity in honey bees (*apis mellifera*)*, Apidologie, 35 (2004), pp. 133–142.
- [20] T. SEELEY, P. VISSCHER, T. SCHLEGEL, P. HOGAN, N. FRANKS, AND J. MARSHALL, *Stop signals provide cross inhibition in collective decision-making by honeybee swarms.*, Science (New York, NY), 335 (2012), p. 108.
- [21] T. D. SEELEY, *The wisdom of the hive: the social physiology of honey bee colonies*, Harvard University Press, 2009.
- [22] ———, *Honeybee Democracy*, Princeton Univ. Press, Princeton, NJ, 2010.
- [23] T. D. SEELEY, S. CAMAZINE, AND J. SNEYD, *Collective decision-making in honey bees: how colonies choose among nectar sources*, Behavioral Ecology and Sociobiology, 28 (1991), pp. 277–290.
- [24] D. SUMPTER AND S. PRATT, *A modelling framework for understanding social insect foraging*, Behavioral Ecology and Sociobiology, 53 (2003), pp. 131–144.
- [25] P. TENCZAR, C. C. LUTZ, V. D. RAO, N. GOLDENFELD, AND G. E. ROBINSON, *Automated monitoring reveals extreme interindividual variation and plasticity in honeybee foraging activity levels*, Animal Behaviour, 95 (2014), pp. 41–48.
- [26] K. VON FRISCH, *The Dance Language and Orientation of Bees*, Cambridge University Press, 1967.
- [27] K. VON FRISCH, *Bees: Their vision, Chemical Senses, and Language*, Cornell University Press, 1971.

Appendix B

November 6, 2019

Royal Society Open Science
Editorial Board

Dear Editors,

We thank the referees again for their feedback on our manuscript “Social inhibition maintains adaptivity and consensus of honey bees foraging in dynamic environments.” We have addressed the remaining minor comments from Reviewers 2 and 3 in this revision. Details are given below, and we have highlighted changes in the manuscript in blue.

We hope our manuscript can now be published in Royal Society Open Science.

Zachary P. Kilpatrick, PhD
Assistant Professor
Department of Applied Mathematics
University of Colorado Boulder
email: zpkilpat@colorado.edu

Reviewers' & editor's comments are italicized. Our responses are in plain text. Changes to the manuscript text are in blue.

Reviewer 2: thanks you very much for your detailed and appropriate responses to my concerns.

One minor thing, you might check the following paper and cite it if you agree that it is appropriate for your manuscript: Townsend-Mehler, J. M., & Dyer, F. C. (2011). An integrated look at decision-making in bees as they abandon a depleted food source. Behavioral Ecology and Sociobiology, 66(2), 275-286.

Thanks for pointing us to this paper, we now reference it when discussing bees' ability to abandon a low quality feeder in favor of one that is higher quality.

Reviewer 3: Second, I disagree with several of reviewer 2's criticisms (notwithstanding correct usage of terminology, which I agree with). The authors' responses are sound in my opinion. In particular if recruits and scouts are distinct castes then there is no scope for positive feedback to good quality nest / forage sites, since recruits will not recruit, and scouts will not follow. Hence models over the previous several decades are typically considering these as distinct behavioural states, as the authors respond. These models usually implicitly or explicitly consider only the forager class of bees (a small fraction of the total). Great, we are glad to be in agreement with you.

I like this work but have a general criticism around the objective function assumed; reward rate is probably fine as a fast-timescale criterion, but I think the lack of crowding in the reward rate function needs to be addressed. As it stands, the simple linear dependence of intake on proportion of scouts is intuitive and straightforward to see, apart from the need to

adapt to changes in forage patch quality. Yet the authors repeatedly refer to crowding in discussing the natural history. Foraging models with crowding can yield much more complex optimal strategies, even with static options; see, for example, Talamali et al. (<https://doi.org/10.1007/s11721-019-00176-9> - Fig 10). I can see two solutions for the authors: (1) do a lot more work including redoing the optimality analysis, (2) acknowledge that crowding is neglected and motivate this, e.g. by a very-large-forage patch assumption.

Thank you for pointing this out. We now acknowledge the ignorance of the crowding effect when introducing the model: We assume feeders are large enough to accommodate all the bees in the colony and hence we neglect the effect of crowding. Foraging efficacy is thus quantified by the group reward rate (RR), assuming net nutrition is proportional to the fraction of the colony at a feeder u_X times the current quality of that feeder minus the foraging cost c (e.g., energy required to forage), $\alpha_X(t) - c$.

and we have added a sentence on possible extensions in the Discussion

We could also have considered the effects of crowding at feeders [1], so nutrition yields would scale sublinearly with the fraction of bees at the feeder, possibly reordering the efficacy of social inhibition strategies.

I think neglected some relevant results in one of the references they cite. Pais et al. [10] analysed the dynamics of their stop-signal model in a periodically changing environment - they showed a hysteresis loop for this model.

We now note: Such hysteresis in stop-signaling populations was also identified in [2].

Regarding the result that indiscriminate stop-signalling does not admit bistability, it is worth noting that a similar analysis was done for the model of Seeley et al. (2012) in the supplementary material for that paper.

We have added a reference pointing readers to the Supporting material of Seeley et al. (2012) for more information.

Minor comments:

Introduction

p.1, l.50: I don't think Seeley et al. interpreted the stop-signalling as dissuading recipients of signals from commitment to 'less suitable nest sites', since signalling was assumed to be targeted but symmetric; thus it was simply targeting individuals with different information.

We have replaced 'less suitable nest sites' to 'overexploited sources' and modified the citation.

p.2, l.9: I don't think nest sites change at a relevant timescale for decision-making... site qualities will change between swarming events..

footnote 3: I don't think consideration of evidence-weighting in collectives is that novel; abandonment implements leak from recruiter populations which is well known as implementing a weighted average for neural accumulators in neuroscience, and considered similarly by some social insect modellers..

We have removed these suggestive sentences and references and have thus shortened the paragraph to conclude with:

We propose that inhibitory social interactions are important for foraging groups to adapt to change in a fluid world.

Discussion

p.11, l.43: I think ref [29] should also be added to the references here, as the form of models presented there inspired several subsequent studies.

Good point, we have added the reference there.

References

- [1] C. W. LAU AND J. C. NIEH, *Honey bee stop-signal production: temporal distribution and effect of feeder crowding*, *Apidologie*, 41 (2010), pp. 87–95.
- [2] D. PAIS, P. M. HOGAN, T. SCHLEGEL, N. R. FRANKS, N. E. LEONARD, AND J. A. MARSHALL, *A mechanism for value-sensitive decision-making*, *PloS one*, 8 (2013), p. e73216.
- [3] T. SEELEY, P. VISSCHER, T. SCHLEGEL, P. HOGAN, N. FRANKS, AND J. MARSHALL, *Stop signals provide cross inhibition in collective decision-making by honeybee swarms.*, *Science (New York, NY)*, 335 (2012), p. 108.